# Optimized Deferral for Imbalanced Settings

**Corinna Cortes** [1]   **Anqi Mao** [2]   **Mehryar Mohri** [1,2]   **Yutao Zhong** [1]

## Abstract

Learning algorithms can be significantly improved by routing complex or uncertain inputs to specialized experts, balancing accuracy with computational cost. This approach, known as *learning to defer*, is essential in domains like natural language generation, medical diagnosis, and computer vision, where an effective deferral can reduce errors at low extra resource consumption. However, the two-stage learning to defer setting, which leverages existing predictors such as a collection of LLMs or other classifiers, often faces challenges due to an expert imbalance problem. This imbalance can lead to suboptimal performance, with deferral algorithms favoring the majority expert. We present a comprehensive study of two-stage learning to defer in expert imbalance settings. We cast the deferral loss optimization as a novel cost-sensitive learning problem over the input-expert domain. We derive new margin-based loss functions and guarantees tailored to this setting, and develop novel algorithms for cost-sensitive learning. Leveraging these results, we design principled deferral algorithms, MILD (*Margin-based Imbalanced Learning to Defer*), specifically suited for expert imbalance settings. Extensive experiments demonstrate the effectiveness of our approach, showing clear improvements over existing baselines on both image classification and real-world Large Language Model (LLM) routing tasks.

## 1. Introduction

Effective learning algorithms often benefit from routing complex or ambiguous inputs to specialized experts. These experts may be particularly effective in handling intricate domains with overlapping class boundaries or in resolving uncertainty near decision thresholds. They can range from human specialists with deep domain expertise to sophisticated yet computationally intensive machine learning models. The challenge lies in dynamically assigning each input to the most suitable expert while balancing the trade-off between accuracy and computational cost. This becomes even more involved when working with a diverse set of experts, each with distinct capabilities and limitations.

The fundamental challenge is to learn from labeled examples to route input instances to the most appropriate experts, a problem known as that of *learning to defer with multiple experts*. In natural language generation, tackling this issue is essential for reducing errors and hallucinations and improving the efficiency of large language models (LLMs) (Wei et al., 2022; Bubeck et al., 2023). Beyond NLP, expert selection strategies are equally critical in domains such as medical diagnosis, image annotation, economic forecasting, and computer vision, where selecting the right expert can significantly enhance both accuracy and resource efficiency.

The *single-stage learning to defer* paradigm has been extensively studied, beginning with foundational research on learning with abstention by Cortes et al. (2016a;b; 2024a), and followed by extensive work on abstention and deferral (Madras et al., 2018; Raghu et al., 2019; Mozannar & Sontag, 2020; Wilder et al., 2021; Pradier et al., 2021; Keswani et al., 2021; Raman & Yee, 2021; Liu et al., 2022; Verma & Nalisnick, 2022; Charusaie et al., 2022; Cao et al., 2022; Verma et al., 2023; Mao et al., 2024a;b;c;h; Mozannar et al., 2023). In this single-stage approach, a predictor and a deferral function are learned jointly, with the deferral function determining the best expert for each input.

However, in many practical scenarios, strong predictors, such as a family of LLMs, are already available, and retraining them alongside a deferral function can be computationally prohibitive. In addition, the models, whether LLMs or other classifiers, may be trained on privacy-sensitive data and only the final models are available for general use. Thus, the single-stage learning to defer framework and its associated methods often overlook the practical constraints encountered in real-world applications. To address these limitations, Mao et al. (2023a) introduced and studied

[1]Google Research, New York, NY; [2]Courant Institute of Mathematical Sciences, New York, NY. Correspondence to: Corinna Cortes <corinna@google.com>, Anqi Mao <aqmao@cims.nyu.edu>, Mehryar Mohri <mohri@google.com>, Yutao Zhong <yutaozhong@google.com>.

*Proceedings of the 43$^{rd}$ International Conference on Machine Learning*, Seoul, South Korea. PMLR 306, 2026. Copyright 2026 by the author(s).

the *two-stage learning to defer* framework, where the family of predictors is fixed and only the deferral function is learned. They provided non-asymptotic learning guarantees and effective algorithms, demonstrating strong empirical performance.

Nevertheless, learning to defer often faces a significant additional challenge: in many real-world settings, a small subset of experts is disproportionately favored across most instances, resulting in *expert imbalance*. This issue arises in various contexts, including LLM-based deferral (Mohri et al., 2024) and top-$k$ prediction tasks (Cortes et al., 2024b). As a consequence, deferral algorithms may overlook less frequently applicable, highly specialized, though possibly costly, experts, and as a result perform only marginally better than naïve baselines that default to the majority expert, see Appendix B for a further discussion and shortcoming of current approaches. Thus, imbalance may hinder the learning process and can lead to a degraded overall performance with only small resource gains. Can we design new imbalance-aware deferral algorithms that outperform the best existing methods, yet keep resource needs low?

In the multi-class setting, a common strategy for addressing imbalance involves oversampling underrepresented classes or undersampling dominant ones (Chawla et al., 2002; Wallace et al., 2011; Kubat & Matwin, 1997; Qiao & Liu, 2009; Han et al., 2005; Estabrooks et al., 2004; Liu et al., 2008; Zhang & Pfister, 2021). Another related approach assigns different loss penalties to different classes, in the hope of making the learning algorithm select the specialized experts more often (see Appendix A). However, these methods lack strong theoretical justification, as they modify the training distribution to diverge from the true target distribution. Empirically, their effectiveness is inconsistent and often depends on extensive hyperparameter tuning (Van Hulse et al., 2007). In the deferral setting, such techniques are even more problematic since they would require assigning additional costs to experts, while the deferral problem already incorporates instance-specific expert costs.

A further challenge unique to the deferral problem is that the learning distribution is defined over input-label pairs, whereas the imbalance we aim to correct concerns experts, not class labels. A 1-D example is provided in Figure 1 illustrating in colors the densities of 4 classes and accuracies of 3 experts, depicted in shades of gray. The cost of the experts varies, with the darkest gray representing the most accurate but also the most expensive expert. In this setting, the left-most expert achieves the highest accuracy over most of the input distribution. The task of the two-stage deferral learning problem is to determine which regions of the input should be deferred to which experts to achieve an optimal trade-off between cost and accuracy. As illustrated in the figure, this choice may be independent of class labels. This distinction between labels and experts complicates the direct application of traditional imbalance-handling techniques. Can we design a principled algorithm for deferral that effectively accounts for expert imbalance while preserving theoretical soundness and guarantees?

**Our contributions.** We present a detailed study of two-stage learning to defer in imbalanced settings. We frame deferral loss minimization as a cost-sensitive learning problem over the input-expert domain, introducing a new distribution over input-expert pairs (Section 3). This leads us to study cost-sensitive multi-class classification under imbalance. In Section 4, we build on recent margin-based methods to introduce new cost-sensitive margin losses and establish guarantees tailored for imbalance. We propose novel algorithms grounded in this theory, which also improve solutions in balanced structured prediction tasks. In Section 5, we design deferral algorithms leveraging class-independent cost structures, prove strong hypothesis-dependent consistency guarantees, and unify input-expert optimization with standard input-label frameworks. Finally, in Section 6, we evaluate MILD against the baseline (Mao et al., 2023a) across CIFAR-10, CIFAR-100, SVHN, and Tiny ImageNet. We further validate our framework on a real-world LLM routing task (MMLU), showing that MILD effectively reduces computational costs by deferring to lightweight models (0.5B/1.5B) when appropriate, thereby overcoming the tendency of baselines to overuse expensive generalist models.

## 2. Preliminaries

We consider a standard multi-class classification setting, with an input space $\mathcal{X}$ and a label space $\mathcal{Y} = [c] \coloneqq \{1, \ldots, c\}$, where $c$ is the number of classes. We consider the general stochastic setting where a joint distribution $\mathcal{D}$ over $\mathcal{X} \times \mathcal{Y}$ allows each $x$ to have multiple possible labels. The expectation $\mathbb{E}_{y|x}$ is taken over the conditional distribution $\mathcal{D}_{y|x}$, which introduces randomness into the problem.

We study the *two-stage learning to defer* framework with multiple experts. In this setting, we are given a set of $p \geq 2$ experts, $g_1, \ldots, g_p$. In our setting, all experts are fixed pre-

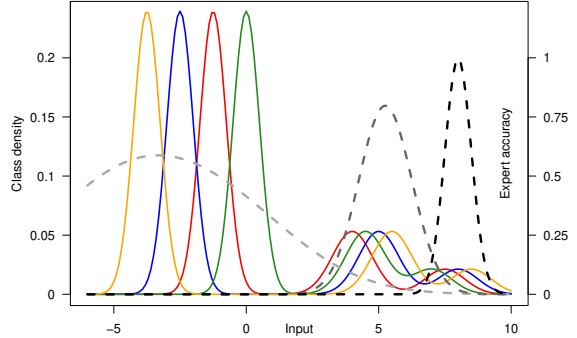

*Figure 1.* 1-D example with 4 classes (colored densities) and 3 experts (gray lines indicating accuracies). Experts have increasing costs indicated by darker shades of gray.

trained models, and only the router $f$ is learned. Each expert is represented as a function mapping $\mathcal{X} \times \mathcal{Y}$ to $\mathbb{R}$. The learner's goal is to select the most suitable expert $g_k$ for each input, balancing expert accuracy and inference cost.

Formally, let $\mathcal{F}$ be a hypothesis set of functions mapping $\mathcal{X} \times [p]$ to $\mathbb{R}$, and let $\mathcal{F}_{\text{all}}$ be the set of all measurable functions with the same domain and range. The goal is to find a predictor $f \in \mathcal{F}$ that minimizes the *deferral loss function*, $\mathsf{L}_{\text{def}}$, defined for any $f \in \mathcal{F}_{\text{all}}$ and $(x, y) \in \mathcal{X} \times \mathcal{Y}$ by:

$$\mathsf{L}_{\text{def}}(f, x, y) = \sum_{k=1}^{p} c_k(x, y) 1_{\mathsf{f}(x)=k},$$

where $\mathsf{f}(x) = \operatorname{argmax}_{k \in [p]} f(x, k)$ represents the prediction of an expert by $f$ for input $x$, using the highest index for tie-breaking. The cost $c_k(x, y)$ is incurred when expert $g_k$ is selected.

A common choice for $c_k$ incorporates $g_k$'s classification error and inference cost (typically computational) (Mao et al., 2023a); for example, $c_k(x, y) = 1_{\mathsf{g}_k(x) \neq y} + \beta_k$, where $\mathsf{g}_k(x) = \operatorname{argmax}_{y \in [c]} g_k(x, y)$ represents the prediction made by expert $g_k$ for input $x$ and $\beta_k$ accounts for the inference cost of expert $g_k$.

The deferral generalization error of a hypothesis $f$, denoted by $\mathcal{E}_{\mathsf{L}_{\text{def}}}(f)$, is the expected deferral loss of $f$ over the data distribution $\mathcal{D}$: $\mathcal{E}_{\mathsf{L}_{\text{def}}}(f) = \mathbb{E}_{(x,y) \sim \mathcal{D}}[\mathsf{L}_{\text{def}}(f, x, y)]$. The best-in-class generalization error of $\mathcal{F}$, denoted by $\mathcal{E}_{\mathsf{L}_{\text{def}}}^*(\mathcal{F})$, is the infimum of the generalization errors over all hypotheses in $\mathcal{F}$: $\mathcal{E}_{\mathsf{L}_{\text{def}}}^*(\mathcal{F}) = \inf_{f \in \mathcal{F}} \mathcal{E}_{\mathsf{L}_{\text{def}}}(f)$. We will adopt similar definitions for other loss functions.

## 3. Deferral Loss Function

In this section, we formulate the minimization of the deferral loss as a cost-sensitive learning problem over the input-expert domain. To achieve this, we first derive alternative expressions for $\mathsf{L}_{\text{def}}$. Recall that the *margin of* $f \in \mathcal{F}$ for the labeled pair $(x, k) \in \mathcal{X} \times [p]$ is defined as $\rho_f(x, k) = f(x, k) - \max_{k' \neq k} f(x, k')$, which represents the difference of the score assigned by $f$ to the pair $(x, k)$ and that of the runner-up expert. The following lemma provides a margin-based reformulation of the deferral loss.

**Lemma 3.1.** *For any $f \in \mathcal{F}$ and $(x, y) \in \mathcal{X} \times \mathcal{Y}$, the loss function $\mathsf{L}_{\text{def}}$ can be expressed as follows:*

$$\mathsf{L}_{\text{def}}(f, x, y)$$
$$= \sum_{k=1}^{p} \left( \sum_{k'=1}^{p} c_{k'}(x, y) 1_{k' \neq k} \right) 1_{\rho_f(x,k) \leq 0} - (p-2) \sum_{k=1}^{p} c_k(x, y).$$

Assuming that the costs satisfy $c_k \in [0, 1]$, which can be achieved through appropriate normalization, then the deferral loss function takes the following general form.

**Lemma 3.2.** *For any $f \in \mathcal{F}$ and $(x, y) \in \mathcal{X} \times \mathcal{Y}$, the loss function $\mathsf{L}_{\text{def}}$ can be expressed as follows:*

$$\mathsf{L}_{\text{def}}(f, x, y)$$
$$= \sum_{k=1}^{p} (1 - c_k(x, y)) 1_{\rho_f(x,k) \leq 0} + \sum_{k=1}^{p} c_k(x, y) - (p-1).$$

The proofs for both lemmas are presented in Appendix E. Based on these results, and ignoring constants that do not depend on $f$, the loss $\mathsf{L}_{\text{def}}$ can be equivalently written as:

$$\forall (f, x, y), \quad \mathsf{L}_{\text{def}}(f, x, y) = \sum_{k=1}^{p} \overline{c}_k(x, y) 1_{\rho_f(x,k) \leq 0},$$

for appropriate *rewards* $\overline{c}_k$ (with $k \in [p]$), which vary over the support of the joint distribution $\mathcal{D}$. In particular, Lemma 3.1 leads to an equivalent form of $\mathsf{L}_{\text{def}}$ by setting reward $\overline{c}_k(x, y) = \sum_{k' \neq k} c_{k'}(x, y)$ and Lemma 3.2 yields an alternative equivalent form with reward $\overline{c}_k(x, y) = 1 - c_k(x, y)$.

Fix $x \in \mathcal{X}$ and define $\mathsf{p}(k|x) = \frac{\mathbb{E}_{y|x}[\overline{c}_k(x,y)]}{C(x)}$, which normalizes the reward into a conditional probability, where $C(x) = \mathbb{E}_{y|x}\left[\sum_{k=1}^{p} \overline{c}_k(x, y)\right]$ is the normalization factor. Then, we have

$$\mathbb{E}_{y|x}[\mathsf{L}_{\text{def}}(f, x, y)] = \sum_{k=1}^{p} \mathbb{E}_{y|x}[\overline{c}_k(x, y)] 1_{\rho_f(x,k) \leq 0}$$
$$= C(x) \sum_{k=1}^{p} \mathsf{p}(k|x) 1_{\rho_f(x,k) \leq 0}. \quad (1)$$

Note that the reward $\overline{c}_k$, and consequently the probability $\mathsf{p}(k \mid x)$, is inversely proportional to the cost $c_k$. This aligns with the intuition that samples should be deferred to experts with lower cost. Let $\overline{\mathcal{D}}$ denote the marginal distribution over $\mathcal{X}$ and define the distribution $\mathcal{P}$ over $\mathcal{X} \times [p]$ by $\forall (x, k) \in \mathcal{X} \times [p], \mathcal{P}(x, k) = \mathsf{p}(k|x) \overline{\mathcal{D}}(x)$. Then, we can express the expected deferral loss as:

$$\mathbb{E}_{(x,y) \sim \mathcal{D}}[\mathsf{L}_{\text{def}}(f, x, y)] = \mathbb{E}_{(x,k) \sim \mathcal{P}}[\ell_{\text{def}}(f, x, k)]$$
$$\text{with} \quad \ell_{\text{def}}(f, x, k) = C(x) 1_{\rho_f(x,k) \leq 0} \quad (2)$$

where the loss function $\ell_{\text{def}}$ is defined for all $(f, x, k) \in \mathcal{F}_{\text{all}} \times \mathcal{X} \times [p]$. Thus, $\ell_{\text{def}}$ represents a cost-sensitive loss function over the input-expert domain, suitable for addressing our expert imbalance problem. This motivates the study of cost-sensitive multi-class classification with imbalanced data in the following section.

## 4. Imbalanced Cost-Sensitive Multi-Class Classification

This section gives a theoretical analysis of imbalanced cost-sensitive multi-class classification, leveraging recent work

of Cortes et al. (2025a) on imbalanced margin for multi-class classification. We first extend existing theoretical tools to the cost-sensitive instance-dependent case. Then, we present a theoretical analysis of this framework and derive new algorithms for imbalanced cost-sensitive multi-class classification. Our algorithms are novel, even in the balanced case and can lead to improved structured prediction theory and algorithm design (see Appendix I).

### 4.1. Theoretical Analysis

Let $f: \mathcal{X} \times [p] \to \mathbb{R}$ be a scoring function belonging to the hypothesis set $\mathcal{F}$. We define the cost-sensitive zero-one loss function $\mathsf{L}$ as follows: for all $(f, x, k) \in \mathcal{F} \times \mathcal{X} \times [p]$,

$$\mathsf{L}(f, x, k) = c(x, k, \mathsf{f}(x)) \, 1_{\rho_f(x,k) \leq 0},$$

where $c(x, k, \mathsf{f}(x)) \in [0, 1]$ is a non-negative cost bounded by one that is vanishing when $k = \mathsf{f}(x)$. Note that $\ell_{\text{def}}$ is a special case of $\mathsf{L}$.

**A. Cost-sensitive margin loss functions.** We first introduce new cost-sensitive instance-dependent margin loss functions.

Let $\Phi_\rho: u \mapsto \min(1, \max(0, 1 - u/\rho))$ denote the $\rho$-margin loss function. We can upper-bound the cost-sensitive zero-one loss function $\mathsf{L}$ as follows:

$$\begin{aligned}
\mathsf{L}(f, x, k) &\leq c(x, k, \mathsf{f}(x)) \Phi_\rho(\rho_f(x, k)) \\
&= c(x, k, \mathsf{f}(x)) \Phi_\rho\Big(f(x, k) - \max_{k' \neq k} f(x, k')\Big) \\
&= c(x, k, \mathsf{f}(x)) \Phi_\rho(f(x, k) - f(x, \mathsf{f}(x))) \\
&\leq \max_{k' \in [p]} \{ c(x, k, k') \Phi_\rho(f(x, k) - f(x, k')) \}.
\end{aligned}$$

The second equality follows from the fact that when $k = \mathsf{f}(x)$ we have $c(x, k, \mathsf{f}(x)) = 0$. Otherwise, for $k \neq \mathsf{f}(x)$, the runner-up prediction satisfies $\operatorname{argmax}_{k' \neq k} f(x, k') = \mathsf{f}(x)$.

This analysis motivates the definition of the *cost-sensitive margin loss function* as the function $\mathsf{L}_\rho: \mathcal{F}_{\text{all}} \times \mathcal{X} \times [p] \to \mathbb{R}$, defined as follows, for any fixed $\rho > 0$:

$$\mathsf{L}_\rho(f, x, k) = \max_{k' \in [p]} \{ c(x, k, k') \Phi_\rho(f(x, k) - f(x, k')) \}.$$

Inspired by the analysis of Cortes et al. (2025a) for imbalanced learning in standard multi-class classification, we extend our definition to the imbalanced setting. Given a vector of margin parameters $\rho = [\rho_k]_{k \in [p]}$, we introduce the *imbalanced cost-sensitive margin loss function* as the function $\mathsf{L}_\rho: \mathcal{F}_{\text{all}} \times \mathcal{X} \times [p] \to \mathbb{R}$, defined by: for all $(f, x, k) \in \mathcal{F} \times \mathcal{X} \times [p]$,

$$\mathsf{L}_\rho(f, x, k) = \max_{k' \in [p]} \Big\{ c(x, k, k') \Phi_{\rho_k}(f(x, k) - f(x, k')) \Big\}.$$

**B. Margin bounds.** We now establish a general margin-based generalization bound, which serves as the foundation for deriving new algorithms for imbalanced cost-sensitive classification.

To capture class-specific variations in confidence margins, we adopt the definition of *class-sensitive Rademacher complexity from prior work*, which introduces a distinct confidence margin weight $\rho_i$ for each class $i$. Given non-negative confidence margin parameters $\rho = [\rho_k]_{k \in [p]}$, the empirical class-sensitive Rademacher complexity of $\mathcal{F}$ for a sample $S = (x_1, \ldots, x_m)$ is defined as:

$$\widehat{\mathfrak{R}}_{S, \rho}(\mathcal{F}) = \frac{1}{m} \mathbb{E}_\epsilon \left[ \sup_{f \in \mathcal{F}} \left\{ \sum_{j=1}^p \sum_{i \in S_j} \sum_{k=1}^p \epsilon_{ik} \frac{f(x_i, k)}{\rho_j} \right\} \right],$$

where $S_j$ denotes the subsample of $S$ consisting of points labeled with $j$, with cardinality $m_j = |S_j|$, and $\epsilon = (\epsilon_{ik})_{i,k}$ represents a matrix of independent Rademacher variables $\epsilon_{ik}$s, each uniformly distributed over $\{-1, +1\}$. For any integer $m \geq 1$, the class-sensitive Rademacher complexity of $\mathcal{F}$ is the expectation of $\widehat{\mathfrak{R}}_{S, \rho}(\mathcal{F})$ over all samples $S$ of size $m$: $\mathfrak{R}_{m, \rho}(\mathcal{F}) = \mathbb{E}_{S \sim \mathcal{D}^m}[\widehat{\mathfrak{R}}_{S, \rho}(\mathcal{F})]$.

Using these notions of complexity, we prove the following imbalanced cost-sensitive margin bound.

**Theorem 4.1** (Margin bound for imbalanced cost-sensitive classification)**.** *Let $\mathcal{F}$ be a family of functions mapping from $\mathcal{X} \times [p]$ to $\mathbb{R}$, and fix $\rho = [\rho_k]_{k \in [p]}$. Then, for any $\delta > 0$, with probability at least $1 - \delta$, each of the following inequalities holds for all $f \in \mathcal{F}$:*

$$\mathcal{E}_{\mathsf{L}}(f) \leq \widehat{\mathcal{E}}_{S, \rho}(f) + 4\sqrt{2p} \, \mathfrak{R}_{m, \rho}(\mathcal{F}) + \sqrt{\frac{\log \frac{1}{\delta}}{2m}}$$

$$\mathcal{E}_{\mathsf{L}}(f) \leq \widehat{\mathcal{E}}_{S, \rho}(f) + 4\sqrt{2p} \, \widehat{\mathfrak{R}}_{S, \rho}(\mathcal{F}) + 3\sqrt{\frac{\log \frac{2}{\delta}}{2m}}.$$

Our proof (see Appendix F) is similar to that of Cortes et al. (2025a), modulo our adaptation to the instance-dependent cost-sensitive nature of our notion of margin loss. This formulation requires substantial adaptation and extends the previous work, which is limited to the standard class-imbalanced setting with uniform costs. In particular, we establish novel margin bounds based on a refined upper bound involving a maximum operator and derive new Rademacher complexity bounds for this term using the vector contraction lemma. Our bounds can be generalized to hold uniformly for all $\rho = [\rho_k]_{k \in [p]} \in (0, 1]^p$, at the cost of additional $\log \log$-terms, using standard proof techniques (Mohri et al., 2018, Theorem 5.9). As with standard margin bounds, these learning guarantees suggest a trade-off: Increasing $\rho_k$ reduces the complexity term (second term) but simultaneously increases the empirical imbalanced cost-sensitive margin loss, $\widehat{\mathcal{E}}_{S, \rho}(f)$, by imposing stricter confidence margin requirements. Thus, if $f$ maintains a low empirical imbalanced cost-sensitive margin loss even with relatively large $\rho_k$ values, it admits a strong generalization error guarantee.

## 4.2. Algorithms

The margin guarantees established in the previous section provide a foundation for developing new algorithms. We begin by deriving a more explicit learning guarantee within a broad framework, which we then use to define a general cost-sensitive learning algorithm for imbalanced data.

**A. Explicit upper bounds**. To make these guarantees more explicit, we introduce the following setup. Given a feature mapping $\Phi\colon \mathcal{X} \times [p] \to \mathbb{R}^d$, we can identify $\mathcal{X} \times [p]$ with a subset of $\mathbb{R}^d$, with $\|\Phi(x,k)\| \le \mathsf{X}_k$ for all $x \in \mathcal{X}$ and $\mathsf{X} = \max_{k \in [p]} \mathsf{X}_k$, for some norm $\|\cdot\|$. We assume $\mathcal{F}$ is given by $\mathcal{F} = \left\{ f \in \overline{\mathcal{F}}\colon \|f\|_* \le \overline{\mathsf{F}} \right\}$, for some appropriate norm $\|\cdot\|_*$ on some space $\overline{\mathcal{F}}$ and $\overline{\mathsf{F}} > 0$. This formulation covers a wide range of hypothesis sets, including linear, kernel-based, and neural network models. In particular, it captures the settings of neural networks with weight matrices constrained by a Frobenius norm bound (Cortes et al., 2017; Neyshabur et al., 2015) or a spectral norm complexity constraint relative to reference weight matrices (Bartlett et al., 2017). In all of these cases, the empirical class-sensitive Rademacher complexity can be upper bounded as follows:

$$\widehat{\mathfrak{R}}_{S,\boldsymbol{\rho}}(\mathcal{F}) \le \frac{\sqrt{p}\,\mathsf{F}}{m}\sqrt{\sum_{j=1}^{p} \frac{m_j \mathsf{X}_j^2}{\rho_j^2}} \le \frac{\sqrt{p}\,\mathsf{F}\mathcal{X}}{m}\sqrt{\sum_{j=1}^{p} \frac{m_j}{\rho_j^2}}, \quad (3)$$

where the complexity term $\mathsf{F}$ depends on $\overline{\mathsf{F}}$. Combining this upper bound with Theorem 4.1 yields the following more explicit guarantee.

**Corollary 4.2.** *Fix* $\boldsymbol{\rho} = \left[\rho_k\right]_{k \in [p]}$*, then, for any* $\delta > 0$*, with probability at least* $1 - \delta$ *over the choice of a sample $S$ of size $m$, the following holds for any $f \in \mathcal{F}$:*

$$\mathcal{E}_{\mathsf{L}}(f) \le \widehat{\mathcal{E}}_{S,\boldsymbol{\rho}}(f) + \frac{4\sqrt{2}\,p\mathsf{F}}{m}\sqrt{\sum_{j=1}^{p} \frac{m_j \mathsf{X}_j^2}{\rho_j^2}} + 3\sqrt{\frac{\log\frac{2}{\delta}}{2m}}.$$

As with Theorem 4.1, this bound can be generalized to hold uniformly for all $\boldsymbol{\rho} = \left[\rho_k\right]_{k \in [p]} \in (0,1]^p$, at the cost of additional $\log\log$-terms. This generalized guarantee provides a basis for designing algorithms choosing $f \in \mathcal{F}$ and $\boldsymbol{\rho}$ to minimize the bound.

Let $\Psi$ be a decreasing convex function such that $\Phi_\rho(x) \le \Psi\left(\frac{x}{\rho}\right)$ for all $x \in \mathbb{R}$ and $\rho > 0$. $\Psi$ may be the hinge loss, $\Psi(x) = \max(0, 1-x)$, or any member of the broad family of composition-sum (comp-sum) losses (Mao et al., 2023e) defined by $\Psi(x) = \Phi^\tau(e^{-x})$, with $\Phi^\tau$ for $\tau \ge 0$ given by

$$\Phi^\tau(u) = \begin{cases} \frac{1}{1-\tau}\big((1+u)^{1-\tau} - 1\big) & \tau \ge 0, \tau \ne 1 \\ \log(1+u) & \tau = 1, \end{cases}$$

for all $u \ge 0$. This family includes the logistic loss ($\tau = 1$) and the exponential loss ($\tau = 0$). Using the fact that $\Phi_\rho(t) \le$

$\Psi\left(\frac{t}{\rho}\right)$, the guarantee of Corollary 4.2 and its generalization to a uniform bound can be expressed as: for any $\delta > 0$, with probability at least $1 - \delta$, for all $f \in \mathcal{F}$,

$$\mathcal{E}_{\mathsf{L}}(f) \le \frac{1}{m}\left[\sum_{j=1}^{p} \sum_{i \in S_j} \max_{k' \in [p]}\left\{c(x_i, j, k')\Psi\left(\frac{f(x_i,j) - f(x_i,k')}{\rho_j}\right)\right\}\right]$$
$$+ \frac{4\sqrt{2}\mathsf{F}p}{m}\sqrt{\sum_{j=1}^{p} \frac{m_j \mathsf{X}_j^2}{\rho_j^2}} + O\left(\frac{1}{\sqrt{m}}\right),$$

where the last term accounts for the $\log$-$\log$ terms and the $\delta$-confidence term.

**B. General cost-sensitive algorithm.** Define $\overline{\rho} = \sum_{j=1}^{p} \rho_j$ and $\overline{\mathsf{X}} = \sum_{j=1}^{p} (m_j \mathsf{X}_j^2)^{\frac{1}{3}}$. It is straightforward to show that for constant $\overline{\rho}$, the second term of the bound is minimized when $\frac{\rho_j}{\overline{\rho}} = \frac{(m_j \mathsf{X}_j^2)^{\frac{1}{3}}}{\overline{\mathsf{X}}}$ for all $j$. This leads to the following regularization-based algorithm:

$$\frac{1}{m}\left[\sum_{j=1}^{p} \sum_{i \in S_j} \max_{k' \in [p]}\left\{c(x_i, j, k')\Psi\left(\frac{f(x_i,j) - f(x_i,k')}{\rho_j}\right)\right\}\right]$$
$$+ \min_{f \in \overline{\mathcal{F}}} \lambda\|f\|^2, \quad (4)$$

where $\lambda$ and $\rho_j$s are selected via cross-validation, with $\rho_j$s close to $\frac{(m_j \mathsf{X}_j^2)^{\frac{1}{3}}}{\overline{\mathsf{X}}}\overline{\rho}$. For a large number of classes $p$, we can assign the same $\rho_j$ to classes with smaller representation.

# 5. Algorithms for Expert Imbalance Settings

In this section, we develop deferral algorithms for expert imbalance settings, leveraging the results of the previous section. We first derive simplified cost-sensitive algorithms for the settings where the costs depend only on $x$. Next, we establish a strong hypothesis set-dependent consistency guarantee for the corresponding loss function. This result further justifies the proposed algorithm. Finally, we apply these results to the deferral setting, which leads to a novel deferral algorithm for expert imbalance settings with favorable theoretic guarantees.

## 5.1. Algorithms for Class-Independent Cost-Sensitive Learning

When $c(x, k, k')$ is independent of $(k, k')$ (denoted more simply as $C(x)$), as is the case in the deferral setting (i.e., $\ell_{\text{def}}$ in Eq. (2)), choosing $\Psi$ to be the logistic loss, using the monotonicity of the $\log$ function and upper-bounding the maximum by a sum, we obtain:

$$\mathsf{L}_{\boldsymbol{\rho}}(f, x, k) \le C(x) \log\left[1 + \max_{k' \ne k} \exp\left(\frac{f(x,k') - f(x,k)}{\rho_k}\right)\right]$$
$$\le C(x) \log\left[\sum_{k'=1}^{p} \exp\left(\frac{f(x,k') - f(x,k)}{\rho_k}\right)\right].$$

See Appendix G for detailed derivations and formulations with other choices of $\Phi$, such as the comp-sum loss with $\tau \neq 1$. We consider the imbalanced cost-sensitive surrogate loss $\widetilde{\mathsf{L}}_{\boldsymbol{\rho}}$ defined as:

$$\widetilde{\mathsf{L}}_{\boldsymbol{\rho}}(f, x, k) = C(x) \log\left[\sum_{k'=1}^{p} \exp\left(\frac{f(x, k') - f(x, k)}{\rho_k}\right)\right]$$
$$\coloneqq C(x)\, \widetilde{\ell}_{\boldsymbol{\rho}}(f, x, k). \tag{5}$$

Then, in this special case where $c(x, k, k') = C(x)$, the algorithm presented in the previous section can be re-expressed as follows, with $\rho_j$s chosen via cross-validation and close to $\frac{(m_j \mathsf{X}_j^2)^{\frac{1}{3}}}{\mathsf{X}} \overline{\rho}$:

$$\min_{f \in \mathcal{F}} \lambda \|f\|^2 + \frac{1}{m} \sum_{i \in S} C(x_i)\widetilde{\ell}_{\boldsymbol{\rho}}(f, x_i, k_i). \tag{6}$$

### 5.2. Hypothesis-Set Dependent Consistency Bounds

Given a hypothesis set $\mathcal{F}$, an $\mathcal{F}$-*consistency bound* (Awasthi et al., 2022a;b; Mao et al., 2023e) for a surrogate loss $\mathsf{L}_1$ of a target loss function $\mathsf{L}_2$ is an inequality of the form

$$\forall f \in \mathcal{F},\ \mathcal{E}_{\mathsf{L}_2}(f) - \mathcal{E}_{\mathsf{L}_2}^*(\mathcal{F}) + \mathcal{M}_{\mathsf{L}_1}(\mathcal{F})$$
$$\leq \Gamma\big(\mathcal{E}_{\mathsf{L}_1}(f) - \mathcal{E}_{\mathsf{L}_1}^*(\mathcal{F}) + \mathcal{M}_{\mathsf{L}_1}(\mathcal{F})\big), \tag{7}$$

where $\Gamma \colon \mathbb{R}_+ \to \mathbb{R}_+$ is a non-decreasing concave function with $\Gamma(0) = 0$, and $\mathcal{M}_{\mathsf{L}}(\mathcal{F})$ is the *minimizability gap* for hypothesis set $\mathcal{F}$ and loss function $\mathsf{L}$. The minimizability gap is defined as the difference between the best-in-class expected loss and that of the expected pointwise infimum loss: $\mathcal{M}_{\mathsf{L}}(\mathcal{F}) = \mathcal{E}_{\mathsf{L}}^*(\mathcal{F}) - \mathbb{E}_x\big[\inf_{f \in \mathcal{F}} \mathbb{E}_{y|x}[\mathsf{L}(f, x, k)]\big]$. Due to the super-additivity of the infimum, the minimizability gap is always non-negative. It becomes zero when the best-in-class error $\mathcal{E}_{\mathsf{L}}^*(\mathcal{F})$ equals the Bayes error $\mathcal{E}_{\mathsf{L}}^*(\mathcal{F}_{\text{all}})$, specifically when $\mathcal{F} = \mathcal{F}_{\text{all}}$ (Mao et al., 2024g). The $\mathcal{F}$-consistency bound relates the minimization of the estimation error for the surrogate loss $\mathsf{L}_1$ to that of the target loss $\mathsf{L}_2$ quantitatively. It provides a stronger and more informative guarantee than Bayes-consistency (Zhang, 2004; Bartlett et al., 2006; Steinwart, 2007), which it implies (by setting $\mathcal{F} = \mathcal{F}_{\text{all}}$). Bayes-consistency is a fundamental guarantee in the study of surrogate losses, including in learning to defer settings (Mozannar & Sontag, 2020; Verma & Nalisnick, 2022), where it ensures that minimizing the excess error of a surrogate loss also minimizes that of the target deferral loss. However, it can be uninformative in practice, as it applies to all measurable functions and ignores the constraints of restricted hypothesis classes. Recent work by Mao et al. (2023a) studies $\mathcal{F}$-consistency bounds for learning to defer, which are more informative because they are specific to the hypothesis class and non-asymptotic. Note that we use the term $\mathcal{F}$-consistency as our hypothesis set is denoted by $\mathcal{F}$.

The following result establishes an $\mathcal{F}$-consistency bound for the imbalanced cost-sensitive surrogate loss $\widetilde{\mathsf{L}}_{\boldsymbol{\rho}}$ introduced

with respect to the cost-sensitive zero-one loss. A hypothesis set $\mathcal{F}$ is considered complete if, for every input-expert pair $(x, k) \in \mathcal{X} \times [p]$, the set of scores $\{f(x, k) \colon f \in \mathcal{F}\}$ spans all real numbers. Most commonly used hypothesis sets are complete.

**Theorem 5.1** ($\mathcal{F}$-consistency bound for imbalanced cost-sensitive surrogate loss). *Let $\mathcal{F}$ be a complete hypothesis set. Then, for all $f \in \mathcal{F}$, $\boldsymbol{\rho} > \mathbf{0}$, the following inequality holds:*

$$\mathcal{E}_{\mathsf{L}}(f) - \mathcal{E}_{\mathsf{L}}^*(\mathcal{F}) + \mathcal{M}_{\mathsf{L}}(\mathcal{F})$$
$$\leq \sqrt{2}\Big(\mathcal{E}_{\widetilde{\mathsf{L}}_{\boldsymbol{\rho}}}(f) - \mathcal{E}_{\widetilde{\mathsf{L}}_{\boldsymbol{\rho}}}^*(\mathcal{F}) + \mathcal{M}_{\widetilde{\mathsf{L}}_{\boldsymbol{\rho}}}(\mathcal{F})\Big)^{\frac{1}{2}}.$$

The proof can be found in Appendix H. Theorem 5.1 provides the first $\mathcal{F}$-consistency guarantee for a cost-sensitive surrogate loss. Even in the standard imbalanced case where the cost $C(x) \equiv 1$, it offers new guarantees for the surrogate loss studied in (Cortes et al., 2025a). In this special case, Theorem 5.1 also extends the standard $\mathcal{F}$-consistency guarantees in (Mao et al., 2023e) to the imbalanced setting.

### 5.3. Application to Deferral

In the context of deferral, to be precise, we should choose $c(x, j, k') = C(x)1_{j=k'}$. However, since $\{j = k'\}$ leads to a constant term, we instead consider the surrogate loss $\widetilde{\mathsf{L}}_{\boldsymbol{\rho}}$ with $\Psi$ chosen as the logistic loss. Alternatively, we can use other functions $\Psi$, such as the comp-sum loss (see Appendix G). By reformulating the input-expert problem within the input-label domain, we obtain:

$$\mathbb{E}_{(x,k) \sim \mathcal{P}}\big[\widetilde{\mathsf{L}}_{\boldsymbol{\rho}}(f, x, k)\big] = \mathbb{E}_{x \sim \mathcal{D}_{\mathcal{X}}}\left[\mathbb{E}_{k \sim \mathsf{p}(\cdot|x)}\big[C(x)\, \widetilde{\ell}_{\boldsymbol{\rho}}(f, x, k)\big]\right]$$
$$= \mathbb{E}_{x \sim \mathcal{D}_{\mathcal{X}}}\left[\sum_{k=1}^{p} C(x)\mathsf{p}(k|x)\widetilde{\ell}_{\boldsymbol{\rho}}(f, x, k)\right]$$
$$= \mathbb{E}_{x \sim \mathcal{D}_{\mathcal{X}}}\left[\sum_{k=1}^{p} \mathbb{E}_{y|x}\big[\overline{c}_k(x, y)\big]\widetilde{\ell}_{\boldsymbol{\rho}}(f, x, k)\right]$$
$$= \mathbb{E}_{(x,y) \sim \mathcal{D}}\left[\sum_{k=1}^{p} \overline{c}_k(x, y)\, \widetilde{\ell}_{\boldsymbol{\rho}}(f, x, k)\right].$$

Thus, given a sample $S$ drawn from $\mathcal{D}^m$, the empirical objective to minimize for our algorithm is $\mathbb{E}_{(x,y) \sim S}\big[\sum_{k=1}^{p} \overline{c}_k(x, y)\, \widetilde{\ell}_{\boldsymbol{\rho}}(f, x, k)\big]$. This leads to a novel algorithm for deferral with expert imbalance, defined by:

$$\min_{f \in \mathcal{F}} \lambda \|f\|^2 + \frac{1}{m} \sum_{i=1}^{m} \sum_{k=1}^{p} \overline{c}_k(x_i, y_i)\widetilde{\ell}_{\boldsymbol{\rho}}(f, x_i, k),$$
$$\text{with } \widetilde{\ell}_{\boldsymbol{\rho}}(f, x, k) = \log\left[\sum_{k'=1}^{p} \exp\left(\frac{f(x, k') - f(x, k)}{\rho_k}\right)\right].$$

Let $\widetilde{\mathsf{L}}_{\text{def}, \boldsymbol{\rho}}(f, x, y) = \sum_{k=1}^{p} \overline{c}_k(x, y)\, \widetilde{\ell}_{\boldsymbol{\rho}}(f, x, k)$ be the corresponding deferral surrogate loss. We call our new algorithm MILD (*Margin-based Imbalanced Learning to Defer*).

*Table 1.* **Synthetic experts.** Comparison of our MILD algorithm with TDEF on CIFAR-10, CIFAR-100, SVHN and Tiny ImageNet: **(a) First cost type** (error rate), **(b) Second cost type** (error rate + cost).

(a)

| Method | Setup | Dataset | Deferral Loss DL~ error rate | Ratio of Expert Deferral (%) 1 | 2 | 3 | 4 | 5 |
|---|---|---|---|---|---|---|---|---|
| TDEF | | CIFAR-10 | 0.0520 ± 0.0022 | 71.31 | 19.18 | 9.51 | — | — |
| MILD | | | **0.0403 ± 0.0018** | 70.01 | 19.99 | 10.00 | — | — |
| TDEF | | CIFAR-100 | 0.2399 ± 0.0019 | 83.26 | 12.54 | 4.20 | — | — |
| MILD | (I) | | **0.2272 ± 0.0037** | 81.21 | 12.76 | 6.03 | — | — |
| TDEF | | SVHN | 0.0468 ± 0.0015 | 83.27 | 12.01 | 4.72 | — | — |
| MILD | | | **0.0254 ± 0.0016** | 80.33 | 13.39 | 6.28 | — | — |
| TDEF | | Tiny ImageNet | 0.3488 ± 0.0028 | 71.80 | 28.16 | 0.04 | — | — |
| MILD | | | **0.3365 ± 0.0033** | 70.68 | 19.25 | 10.07 | — | — |
| TDEF | | CIFAR-10 | 0.0924 ± 0.0046 | 51.54 | 19.53 | 18.76 | 10.17 | — |
| MILD | | | **0.0847 ± 0.0038** | 51.50 | 19.41 | 18.99 | 10.10 | — |
| TDEF | | CIFAR-100 | 0.2982 ± 0.0028 | 53.71 | 18.32 | 20.54 | 7.43 | — |
| MILD | (II) | | **0.2899 ± 0.0019** | 55.21 | 17.78 | 18.53 | 8.48 | — |
| TDEF | | SVHN | 0.0604 ± 0.0027 | 63.47 | 14.87 | 14.76 | 6.91 | — |
| MILD | | | **0.0342 ± 0.0018** | 63.99 | 14.56 | 14.07 | 7.38 | — |
| TDEF | | Tiny ImageNet | 0.5287 ± 0.0032 | 47.81 | 35.10 | 14.78 | 2.31 | — |
| MILD | | | **0.5072 ± 0.0036** | 54.69 | 12.23 | 14.82 | 18.26 | — |
| TDEF | | CIFAR-10 | 0.1062 ± 0.0017 | 38.94 | 20.93 | 20.90 | 9.87 | 9.36 |
| MILD | | | **0.0903 ± 0.0019** | 40.40 | 20.13 | 19.94 | 10.73 | 8.80 |
| TDEF | | CIFAR-100 | 0.3215 ± 0.0023 | 46.35 | 17.23 | 20.11 | 8.99 | 7.32 |
| MILD | (III) | | **0.3128 ± 0.0032** | 42.69 | 20.64 | 19.32 | 10.02 | 7.33 |
| TDEF | | SVHN | 0.0684 ± 0.0019 | 53.47 | 17.47 | 15.75 | 7.72 | 5.59 |
| MILD | | | **0.0353 ± 0.0020** | 54.12 | 18.46 | 14.75 | 6.17 | 6.50 |
| TDEF | | Tiny ImageNet | 0.5857 ± 0.0038 | 36.38 | 17.02 | 36.02 | 9.95 | 0.63 |
| MILD | | | **0.5656 ± 0.0029** | 46.09 | 18.91 | 31.29 | 3.24 | 0.47 |

(b)

| Method | Setup | Dataset | Deferral Loss DL~ error + cost | Ratio of Expert Deferral (%) 1 | 2 | 3 | 4 | 5 |
|---|---|---|---|---|---|---|---|---|
| TDEF | | CIFAR-10 | 0.5950 ± 0.0011 | 63.68 | 23.61 | 12.71 | — | — |
| MILD | | | **0.5779 ± 0.0018** | 67.02 | 20.04 | 12.94 | — | — |
| TDEF | | CIFAR-100 | 0.8150 ± 0.0037 | 57.13 | 23.66 | 19.21 | — | — |
| MILD | (I) | | **0.7928 ± 0.0032** | 50.43 | 27.27 | 22.30 | — | — |
| TDEF | | SVHN | 0.6285 ± 0.0026 | 77.75 | 14.71 | 7.54 | — | — |
| MILD | | | **0.6170 ± 0.0024** | 78.33 | 14.58 | 7.09 | — | — |
| TDEF | | Tiny ImageNet | 0.8819 ± 0.0016 | 4.57 | 75.31 | 20.12 | — | — |
| MILD | | | **0.8653 ± 0.0019** | 8.49 | 43.39 | 48.12 | — | — |
| TDEF | | CIFAR-10 | 0.4421 ± 0.0034 | 48.11 | 22.81 | 18.31 | 10.77 | — |
| MILD | | | **0.4240 ± 0.0021** | 44.87 | 22.65 | 20.50 | 11.98 | — |
| TDEF | | CIFAR-100 | 0.6687 ± 0.0044 | 42.92 | 23.50 | 23.31 | 10.27 | — |
| MILD | (II) | | **0.6506 ± 0.0032** | 45.04 | 20.86 | 21.04 | 13.06 | — |
| TDEF | | SVHN | 0.4265 ± 0.0017 | 60.79 | 15.50 | 15.85 | 7.86 | — |
| MILD | | | **0.4148 ± 0.0023** | 61.04 | 17.86 | 14.04 | 7.06 | — |
| TDEF | | Tiny ImageNet | 0.8576 ± 0.0028 | 32.27 | 16.68 | 18.72 | 32.33 | — |
| MILD | | | **0.8324 ± 0.0019** | 28.91 | 7.27 | 24.55 | 39.27 | — |
| TDEF | | CIFAR-10 | 0.3684 ± 0.0013 | 37.53 | 22.33 | 18.14 | 11.75 | 10.25 |
| MILD | | | **0.3512 ± 0.0015** | 37.55 | 20.26 | 20.48 | 10.09 | 11.62 |
| TDEF | | CIFAR-100 | 0.6051 ± 0.0055 | 37.46 | 18.63 | 20.67 | 12.77 | 10.47 |
| MILD | (III) | | **0.5859 ± 0.0047** | 32.23 | 21.34 | 21.39 | 13.46 | 11.58 |
| TDEF | | SVHN | 0.3412 ± 0.0031 | 52.13 | 18.02 | 15.93 | 6.39 | 7.53 |
| MILD | | | **0.3290 ± 0.0022** | 52.22 | 19.02 | 15.82 | 6.63 | 6.31 |
| TDEF | | Tiny ImageNet | 0.8481 ± 0.0035 | 33.65 | 42.02 | 8.92 | 8.82 | 6.59 |
| MILD | | | **0.8167 ± 0.0031** | 17.22 | 10.02 | 15.48 | 35.43 | 21.85 |

**Intuitive interpretation.** Intuitively, MILD dynamically adjusts the confidence threshold required to defer to each expert based on their overall utility and cost. By mathematically penalizing the over-selection of a dominant expert via the margin parameter $\rho_k$, MILD forces the router to require exponentially higher confidence to defer to the most expensive or dominant model. This effectively makes it "cheaper" for the router to trust highly specialized or underutilized experts, preventing the system from lazily collapsing to the most common choice.

By reformulating the input-expert problem within the input-label domain, Theorem 5.1 directly yields the following $\mathcal{F}$-consistency bound for the deferral loss.

**Corollary 5.2.** *Let $\mathcal{F}$ be a complete hypothesis set. Then, for all $f \in \mathcal{F}$, $\rho > 0$, the following holds:*

$$\mathcal{E}_{L_{def}}(f) - \mathcal{E}^*_{L_{def}}(\mathcal{F}) + \mathcal{M}_{L_{def}}(\mathcal{F})$$
$$\leq \sqrt{2}\Big(\mathcal{E}_{\widetilde{L}_{def,\rho}}(f) - \mathcal{E}^*_{\widetilde{L}_{def,\rho}}(\mathcal{F}) + \mathcal{M}_{\widetilde{L}_{def,\rho}}(\mathcal{F})\Big)^{\frac{1}{2}}.$$

As mentioned above, when $\mathcal{E}^*_L(\mathcal{F}) = \mathcal{E}^*_L(\mathcal{F}_{all})$, the minimizability gaps vanish. A special case occurs when $\mathcal{F} = \mathcal{F}_{all}$. Thus, we further obtain the following excess error bound for the deferral loss.

**Corollary 5.3.** *For all $f \in \mathcal{F}$, $\rho > 0$, the following excess error bound holds:*

$$\mathcal{E}_{L_{def}}(f) - \mathcal{E}^*_{L_{def}}(\mathcal{F}_{all}) \leq \sqrt{2}\Big(\mathcal{E}_{\widetilde{L}_{def,\rho}}(f) - \mathcal{E}^*_{\widetilde{L}_{def,\rho}}(\mathcal{F}_{all})\Big)^{\frac{1}{2}}.$$

Corollaries 5.2 and 5.3 provide strong theoretical guarantees for the deferral algorithm based on minimizing the deferral surrogate loss $\widetilde{L}_{def,\rho}$. In particular, when the surrogate estimation loss $\mathcal{E}_{\widetilde{L}_{def,\rho}}(f) - \mathcal{E}^*_{\widetilde{L}_{def,\rho}}(\mathcal{F})$ is reduced to a small value of $\epsilon$, the target deferral estimation loss $\mathcal{E}_{L_{def}}(f) - \mathcal{E}^*_{L_{def}}(\mathcal{F})$ is upper bounded by $\sqrt{2\epsilon}$.

# 6. Experiments

We evaluated MILD against the state-of-the-art baseline TDEF (Mao et al., 2023a) on image classification benchmarks and a Large Language Model (LLM) routing task. TDEF minimizes a surrogate loss shown to be $\mathcal{F}$-consistent and serves as the primary baseline, being the only existing method designed specifically for two-stage multi-expert deferral (see Section 6.2 and Appendix D for the relationship to standard cost-sensitive and confidence-based methods). Both methods are trained using a logistic surrogate loss. For MILD, we define the reward $\bar{c}_k(x, y) = \sum_{k' \neq k} c_{k'}(x, y)$ following Lemma 3.1. Choice of $\rho$ in MILD follows the theoretical optima (see Appendix C.4).

We report *Deferral Loss (DL)*, the average target loss $L_{def}(f, x, y)$ on test data, alongside expert deferral ratios. Results are reported as mean ± standard deviation over five runs. For simplicity, we omit the standard deviations of the deferral ratios.

**Image Classification Benchmarks.** We used CIFAR-10, CIFAR-100 (Krizhevsky, 2009), SVHN (Netzer et al., 2011), and Tiny ImageNet (Le & Yang, 2015). These vision tasks serve as controlled proxies where expert imbalance perfectly mirrors class imbalance, allowing us to mathematically isolate the mechanics of the router. To simulate varying degrees

*Table 2.* **Real experts.** Comparison of our MILD algorithm with TDEF on CIFAR-10, CIFAR-100, SVHN with real experts: **(a) First cost type** (error rate), **(b) Second cost type** (error rate + cost).

(a)

| Method | Setup | Dataset | Deferral Loss DL ~ error rate | Ratio of Expert Deferral (%) 1 | 2 | 3 | 4 | 5 |
|---|---|---|---|---|---|---|---|---|
| TDEF | | CIFAR-10 | 0.1170 ± 0.0027 | 62.49 | 25.77 | 11.74 | — | — |
| MILD | | | **0.1060 ± 0.0020** | 66.87 | 20.75 | 12.38 | — | — |
| TDEF | (I) | CIFAR-100 | 0.4368 ± 0.0059 | 87.64 | 7.30 | 5.06 | — | — |
| MILD | | | **0.4265 ± 0.0042** | 76.73 | 13.31 | 9.96 | — | — |
| TDEF | | SVHN | 0.0752 ± 0.0013 | 51.58 | 26.99 | 21.43 | — | — |
| MILD | | | **0.0579 ± 0.0015** | 77.16 | 17.39 | 5.45 | — | — |
| TDEF | | CIFAR-10 | 0.1480 ± 0.0020 | 46.95 | 18.64 | 23.08 | 11.33 | — |
| MILD | | | **0.1339 ± 0.0023** | 45.53 | 22.17 | 21.57 | 10.73 | — |
| TDEF | (II) | CIFAR-100 | 0.4620 ± 0.0064 | 63.66 | 13.84 | 13.56 | 8.94 | — |
| MILD | | | **0.4375 ± 0.0048** | 56.15 | 18.07 | 15.76 | 10.02 | — |
| TDEF | | SVHN | 0.0831 ± 0.0027 | 32.44 | 23.27 | 26.61 | 17.68 | — |
| MILD | | | **0.0634 ± 0.0011** | 50.63 | 20.55 | 23.86 | 4.96 | — |
| TDEF | | CIFAR-10 | 0.1554 ± 0.0011 | 33.16 | 22.08 | 18.13 | 16.16 | 10.47 |
| MILD | | | **0.1507 ± 0.0027** | 32.92 | 20.27 | 18.58 | 16.97 | 11.26 |
| TDEF | (III) | CIFAR-100 | 0.4554 ± 0.0023 | 50.01 | 16.73 | 20.18 | 7.49 | 5.59 |
| MILD | | | **0.4419 ± 0.0034** | 48.06 | 19.24 | 17.29 | 7.56 | 7.85 |
| TDEF | | SVHN | 0.0886 ± 0.0012 | 27.21 | 19.98 | 20.28 | 11.78 | 20.75 |
| MILD | | | **0.0707 ± 0.0017** | 45.78 | 19.31 | 15.85 | 6.73 | 12.33 |

(b)

| Method | Setup | Dataset | Deferral Loss DL ~ error + cost | Ratio of Expert Deferral (%) 1 | 2 | 3 | 4 | 5 |
|---|---|---|---|---|---|---|---|---|
| TDEF | | CIFAR-10 | 0.5413 ± 0.0036 | 22.93 | 33.11 | 43.96 | — | — |
| MILD | | | **0.5242 ± 0.0038** | 10.09 | 23.20 | 66.71 | — | — |
| TDEF | (I) | CIFAR-100 | 0.9270 ± 0.0084 | 25.27 | 27.45 | 47.28 | — | — |
| MILD | | | **0.8941 ± 0.0042** | 11.70 | 39.13 | 49.18 | — | — |
| TDEF | | SVHN | 0.3259 ± 0.0017 | 2.06 | 12.04 | 85.90 | — | — |
| MILD | | | **0.3200 ± 0.0009** | 5.94 | 23.98 | 70.08 | — | — |
| TDEF | | CIFAR-10 | 0.4538 ± 0.0016 | 26.88 | 19.50 | 23.58 | 30.05 | — |
| MILD | | | **0.4432 ± 0.0022** | 20.90 | 19.41 | 21.12 | 38.57 | — |
| TDEF | (II) | CIFAR-100 | 0.7824 ± 0.0031 | 39.36 | 22.89 | 19.26 | 18.49 | — |
| MILD | | | **0.7743 ± 0.0037** | 30.48 | 22.55 | 22.06 | 24.91 | — |
| TDEF | | SVHN | 0.2972 ± 0.0029 | 11.74 | 28.73 | 12.54 | 46.99 | — |
| MILD | | | **0.2802 ± 0.0027** | 9.88 | 12.26 | 11.40 | 66.46 | — |
| TDEF | | CIFAR-10 | 0.4121 ± 0.0009 | 27.39 | 18.78 | 4.24 | 12.58 | 37.01 |
| MILD | | | **0.4022 ± 0.0011** | 12.72 | 24.22 | 7.99 | 39.51 | 15.56 |
| TDEF | (III) | CIFAR-100 | 0.7781 ± 0.0053 | 44.55 | 16.41 | 14.67 | 12.03 | 12.34 |
| MILD | | | **0.7639 ± 0.0046** | 33.41 | 17.93 | 18.45 | 13.40 | 16.81 |
| TDEF | | SVHN | 0.2844 ± 0.0046 | 6.14 | 7.92 | 18.60 | 43.07 | 24.27 |
| MILD | | | **0.2682 ± 0.0028** | 12.38 | 15.95 | 18.06 | 25.65 | 27.96 |

of expert imbalance, we constructed three setups: *Setup I* (70%-20%-10% coverage), *Setup II* (50%-20%-20%-10%), and *Setup III* (40%-20%-20%-10%-10%). Detailed configurations are provided in Appendix C.1. We tested two cost functions: (a) *error only*: $c_j = 1_{g_j \neq y}$; and (b) *error + cost*: penalizing both error and expert coverage. We adopted a ResNet-18 (He et al., 2016) as the predictor model. Both methods were trained using the Adam optimizer (Kingma & Ba, 2015) with a batch size of $1,024$, weight decay of $1 \times 10^{-3}$, and learning rate of $1 \times 10^{-3}$ for 200 epochs.

**Synthetic Expert Results.** Table 1 presents results for synthetic experts (perfect accuracy on sub-domains). MILD consistently achieves lower deferral loss by allocating experts closer to the optimal distribution. In the cost-sensitive setting (Table 1(b)), MILD effectively shifts mass from the high-cost Expert 1 to lower-cost experts, outperforming TDEF. Note that the higher loss magnitude in setting (b) reflects the average cost of the setting (sum of cost × coverage), like $0.7^2 + 0.2^2 + 0.1^2 = 0.54$ for Setup I.

**Real Expert Results.** Table 2 presents results using experts trained on data subsets (imperfect accuracy). MILD consistently outperforms TDEF across all settings. For example, on CIFAR-100 (Setup I, Error Only), MILD reduces loss from 0.4368 to 0.4265, successfully mitigating the tendency of TDEF to ignore specialized experts. Further experiments with severe expert imbalance are detailed in Appendix C.3, highlighting the benefits of MILD in extreme scenarios.

### 6.1. LLM Routing on MMLU

**LLM Routing on MMLU.** This task serves as our primary natural imbalance benchmark. Here, expert imbalance is entirely independent of class labels; instead, the multiple-choice labels do not dictate expert specialization. The experts' competence varies naturally based on the inherent linguistic complexity of the input features. This natural capability gap causes the 7B model to dominate organically, demonstrating that expert imbalance occurs independently of class labels.

To validate MILD in this context, we evaluate on the MMLU benchmark (Hendrycks et al., 2021) using the Qwen 2.5 family (Yang et al., 2025) as experts: 7B-Instruct (Strong), 1.5B-Instruct (Medium), and 0.5B-Instruct (Tiny). We compare MILD against TDEF and three additional baselines adapted for routing (Standard Cross-Entropy (CE), Class-Weighted CE (CWCE), and LDAM (Cao et al., 2019)) on a subset of Mathematics and History tasks under two settings: (a) *error only*: Costs based purely on error ($c_j = 1_{g_j \neq y}$). Expert 1 is optimal for 82.2% of queries. (b) *error + cost*: Costs include inference penalties ($\boldsymbol{\beta} = [\beta_1, \beta_2, \beta_3] = [1.0, 0.6, 0.1]$). Efficiency incentives make Expert 3 optimal for 56.8% of queries.

For the LLM routing task, we adopted a DeBERTa-v3-xsmall (He et al., 2023) model as the router. Both methods were fine-tuned for 5 epochs with a learning rate of $5 \times 10^{-5}$ using the AdamW optimizer (Loshchilov & Hutter, 2019). Extended details are provided in Appendix C.2.

Table 3 shows the results. In setting (a), TDEF collapses to the majority expert (99.6% usage of Expert 1), ignoring the 17.8% of cases where smaller models suffice. MILD recovers an imbalanced distribution (83.3% Strong, 10.4% Mid) much closer to the *Optimal* oracle (82.2%, 12.8%).

In setting (b), the contrast is sharper. The *Optimal* strategy uses the Strong expert 31.2% of the time. TDEF fails to learn this, collapsing to 0% usage of the Strong expert (selecting only the cheapest). MILD successfully identifies the complex queries, routing 17.9% to the Strong expert and 73.8% to Tiny, achieving a loss significantly lower than TDEF. Furthermore, MILD ($0.813 \pm 0.040$) significantly outperforms

*Table 3.* LLM Routing on MMLU with Qwen 2.5. MILD adapts to the cost structure and tracks the optimal oracle distribution far better than TDEF, which tends to collapse to extremes.

| Setting | Method | Def. Loss (mean ± std) | Ratio of Expert Deferral (%) | | |
|---|---|---|---|---|---|
| | | | 7B (Str) | 1.5B (Mid) | 0.5B (Tiny) |
| (a) Error Only $(\boldsymbol{\beta} = \vec{0})$ | *Optimal* | — | *82.2* | *12.8* | *5.0* |
| | TDEF | $0.438 \pm 0.080$ | 99.6 | 0.4 | 0.0 |
| | **MILD** | $\mathbf{0.425 \pm 0.080}$ | 83.3 | 10.4 | 6.2 |
| (b) Error + Cost $(\boldsymbol{\beta} = [1, .6, .1])$ | *Optimal* | — | *31.2* | *12.0* | *56.8* |
| | TDEF | $0.928 \pm 0.030$ | 0.0 | 0.0 | 100.0 |
| | **MILD** | $\mathbf{0.813 \pm 0.040}$ | 17.9 | 8.3 | 73.8 |

the other tested baselines: Standard CE ($0.985 \pm 0.038$), CWCE ($0.942 \pm 0.045$), and LDAM ($0.915 \pm 0.035$). Traditional heuristics either ignore instance-dependent routing costs entirely or rely strictly on global static frequencies, which are fundamentally insufficient for dynamic, cost-sensitive expert deferral.

Estimating the offline cost matrix via a small, fixed routing training dataset is a one-time, computationally negligible step compared to the massive live inference compute savings achieved by the routing system during deployment. For instance, in our MMLU experiment, the router was trained on just 4,000 queries. Once deployed, routing millions of real-world queries relies purely on a single forward pass through the lightweight router and incurs zero extra expert evaluation overhead. Furthermore, our mathematical formulation is model-agnostic: whether an expert is a 7B or a 70B+ parameter model, the router strictly operates on the inferred numerical cost distributions, guaranteeing mathematically identical scaling dynamics.

### 6.2. Further Analysis and Discussion

**Ablation Study on $\rho$ Selection.** To isolate the empirical contribution of our theoretical derivation, we evaluated MILD on the LLM MMLU routing task (under the "Error + Cost" setting) in three configurations: 1) MILD *(Uniform)*: a naïve baseline setting all $\rho_k = 1/3$. 2) MILD *(Theory)*: strictly un-tuned theoretically derived values ($\rho_k \propto (m_k \mathsf{X}_k^2)^{1/3}$), entirely skipping cross-validation. 3) MILD *(Tuned)*: tuned via a coarse grid search around the theoretical prior. The theoretical initialization alone reduces the test deferral loss ($0.822 \pm 0.042$) compared to uniform margins ($0.884 \pm 0.055$), nearly matching the fully tuned performance ($0.813 \pm 0.040$). This proves the theoretical bound directly drives the algorithm's success, with cross-validation merely providing minor empirical smoothing.

**Robustness to Noisy Cost-Matrix Estimates.** In practice, expert performance characteristics only need to be estimated once offline over a fixed training set. However, to demonstrate that MILD is robust to imperfect or noisy estimations during this offline training phase, we injected severe noise (20% random Gaussian variations) directly into the estimated cost matrix to simulate highly stochastic and unpredictable expert errors (e.g., LLM hallucinations).

Under this 20% noise setting, the test deferral loss of TDEF degraded from $0.928 \pm 0.030$ (clean) to $1.054 \pm 0.045$. In contrast, MILD remained highly stable, with its test deferral loss only slightly changing from $0.813 \pm 0.040$ (clean) to $0.826 \pm 0.041$. Its robust margin-based decision boundaries act as a buffer that absorbs stochastic noise, preventing the router from overfitting to erratic estimation errors.

**Extreme Imbalance in LLM Routing.** To test the limits on our real-world LLM task, we evaluated an extreme $95\%/5\%$ optimal split on the MMLU dataset (under the "Error + Cost" setting) by heavily skewing the inference routing costs to favor the 7B model. Under extreme imbalance, TDEF collapses entirely to 100% usage of the majority expert ($0.950 \pm 0.030$), failing to identify the remaining 5% of cases where smaller models suffice. In contrast, MILD successfully preserves the minority expert routing (94.6% usage for 7B, 5.4% for smaller models) and achieves a strictly lower overall deferral loss of $0.885 \pm 0.032$.

**Standard Cost-Sensitive Methods and Temperature Scaling.** We also emphasize that existing methods for imbalanced cost-sensitive multiclass classification are not directly applicable here. These methods generally aim to correct for imbalances in the *label* distribution. They do not account for the instance-dependent nature of deferral decisions, where expert suitability varies across the input space regardless of the label. For example, as shown in Figure 1, expert accuracy may depend entirely on the input region even when all labels are equally represented.

The critical distinction between MILD and standard temperature scaling or margin methods (such as (Cao et al., 2019)) is that standard methods adjust logits based on global, static class label frequencies for standard 0-1 classification. In our two-stage deferral setting, MILD applies these adjustments to experts while simultaneously incorporating dynamic, instance-dependent expert costs $\bar{c}_k(x, y)$. Deriving an $\mathcal{F}$-consistent margin bound for this setting required substantial theoretical extensions beyond standard frequency-based adjustments. Consequently, standard cost-sensitive baselines (and static frequency-based margins) fail to capture the structure of the deferral problem, whereas the TDEF baseline and MILD are specifically tailored to this setting.

## 7. Conclusion

We presented principled algorithms to address the inherent imbalance commonly encountered in deferral tasks. By reformulating the minimization of the deferral loss as a cost-sensitive multi-class classification problem, we derived algorithms that are both theoretically sound and effective for handling expert imbalance. Our empirical results demonstrate the benefits of these new deferral algorithms in imbalanced settings.

## Impact Statement

This paper presents work whose goal is to advance the field of Machine Learning. There are many potential societal consequences of our work, none of which we feel must be specifically highlighted here.

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

# Contents of Appendix

# A. Related Work

**Learning to defer.** The *single-stage learning to defer* paradigm has been extensively studied, beginning with foundational research on learning with abstention by Cortes et al. (2016a;b; 2024a), and followed by extensive work on abstention and deferral (Madras et al., 2018; Raghu et al., 2019; Mozannar & Sontag, 2020; Wilder et al., 2021; Pradier et al., 2021; Keswani et al., 2021; Raman & Yee, 2021; Liu et al., 2022; Verma & Nalisnick, 2022; Charusaie et al., 2022; Cao et al., 2022; Verma et al., 2023; Mao et al., 2024a;b;c;h; Mozannar et al., 2023). In this single-stage approach, a predictor and a deferral function are learned jointly, with the deferral function determining the best expert for each input. Specifically, in the abstention setting, where the cost function is constant, Cao et al. (2022); Mao et al. (2024c;b) proposed surrogate losses that are Bayes-consistent (Zhang, 2004; Bartlett et al., 2006; Steinwart, 2007). More generally, in the deferral setting, where the cost function depends on both the instance and the label, Mozannar & Sontag (2020); Charusaie et al. (2022); Verma & Nalisnick (2022); Mozannar et al. (2023); Mao et al. (2024h) proposed Bayes-consistent surrogate losses. Furthermore, Verma et al. (2023); Mao et al. (2024a) extended the surrogate losses in (Verma & Nalisnick, 2022; Mozannar & Sontag, 2020) to the multi-expert deferral setting. In the specific case of abstention, the literature on selective classification provides methods for optimizing the generalization error of non-abstained samples under a fixed selection rate (see, for example, (El-Yaniv et al., 2010; Wiener & El-Yaniv, 2011; El-Yaniv & Wiener, 2012; Wiener & El-Yaniv, 2012; 2015; Geifman & El-Yaniv, 2017; 2019)). However, these methods are not directly applicable to the deferral problem due to the presence of label-dependent costs and multiple experts. Extensions to the regression setting have also been studied in (Wiener & El-Yaniv, 2012; Geifman & El-Yaniv, 2019; Jiang et al., 2020; Zaoui et al., 2020; De et al., 2020; Shah et al., 2022; Li et al., 2024b; Cheng et al., 2023) for abstention and in (Mao et al., 2024f) for multi-expert deferral. Additional research has expanded this framework to include deferring to populations (Tailor et al., 2024), adversarial robustness (Montreuil et al., 2026e), top-$k$ learning (Montreuil et al., 2026b), tailored query mechanisms (Montreuil et al., 2026d), and other extensions (Montreuil et al., 2026a;c;f;g).

However, in many practical scenarios, strong predictors, such as a family of LLMs, are already available, and retraining them alongside a deferral function can be computationally prohibitive. Thus, the single-stage learning to defer framework and its associated methods often overlook the practical constraints encountered in real-world applications. To address these limitations, Mao et al. (2023a) introduced and studied the *two-stage learning to defer* framework, where the family of predictors is fixed and only the deferral function is learned. They provided non-asymptotic learning guarantees and effective algorithms, demonstrating strong empirical performance. Specifically, they developed a novel family of surrogate loss functions and algorithms with broad potential application, especially in LLMs and other practical settings. They proved that these surrogate losses satisfy $\mathcal{H}$-consistency bounds (Awasthi et al., 2022a;b; Mao et al., 2023e;f;c;b;d; 2024d;e; 2025c;a;b; Mao, 2025; Zhong, 2025; DeSalvo et al., 2025; Cortes et al., 2026; Mohri & Zhong, 2026a;b;c;d; Mohri et al., 2026), which are non-asymptotic, hypothesis-set-specific upper bounds on the target estimation loss expressed in terms of the surrogate estimation loss. These bounds provide stronger and more informative guarantees than Bayes-consistency (Zhang, 2004; Bartlett et al., 2006; Steinwart, 2007; Mozannar & Sontag, 2020), which only guarantees that minimizing the surrogate loss over all measurable functions asymptotically minimizes the target loss. This approach differs from post-hoc methods (Okati et al., 2021; Narasimhan et al., 2022) in that it can be used with existing predictors trained in the standard classification setting.

**Learning from imbalanced data.** A common strategy for addressing data imbalance involves oversampling underrepresented classes or undersampling dominant ones (Chawla et al., 2002; Wallace et al., 2011; Kubat & Matwin, 1997; Qiao & Liu, 2009; Han et al., 2005; Estabrooks et al., 2004; Liu et al., 2008; Zhang & Pfister, 2021). Another related approach assigns different loss penalties to different classes (Iranmehr et al., 2019; Masnadi-Shirazi & Vasconcelos, 2010; Elkan, 2001; Zhou & Liu, 2005; Zhao et al., 2018; Zhang et al., 2018; 2019; Sun et al., 2007; Fan et al., 2017; Cui et al., 2019; Jamal et al., 2020; Gabidolla et al., 2024). However, these methods lack strong theoretical justification, as they modify the training distribution in ways that diverge from the true target distribution. Empirically, their effectiveness is inconsistent and often depends on extensive hyperparameter tuning (Van Hulse et al., 2007). In the deferral setting, such techniques are even more problematic since they would require assigning additional costs to experts, while the deferral problem already incorporates instance-specific expert costs.

Beyond these data modification and cost-sensitive learning approaches, there have been a variety of techniques in the imbalanced multi-class classification setting, including logistic loss modifications (Lin et al., 2017; Cao et al., 2019; Tan et al., 2020; Jiawei et al., 2020; Hong et al., 2021; Tian et al., 2020; Menon et al., 2021; Khan et al., 2019; Menon et al., 2021; Ye et al., 2020; Kini et al., 2021; Zhu et al., 2023; Wei et al., 2024; Li et al., 2024a; Cortes et al., 2025b), data augmentation (Wang et al., 2021a; Zhu et al., 2024; Liu et al., 2024; Gao et al., 2023), representation learning (Liu et al.,

2019; Cui et al., 2021; Gao et al., 2024; Meng et al., 2023; Han, 2023), decoupled training (Kang et al., 2020; Zhong et al., 2021), classifier design (Tang et al., 2020; Yang et al., 2022; Kasarla et al., 2022; Shi et al., 2024; Du et al., 2024), ensemble learning (Zhou et al., 2020; Xiang et al., 2020; Wang et al., 2021b; Cui et al., 2022; Zhang et al., 2022; Yang et al., 2024), etc. We refer the reader to a recent survey by Zhang et al. (2023) for a more extensive list and details.

It remains an open question how these techniques can be extended and applied to the deferral problem with expert imbalance. A further challenge unique to deferral is that the learning distribution is defined over input-label pairs, whereas the imbalance we aim to correct concerns the distribution of experts, not labels. This distinction complicates the direct application of traditional imbalance-handling techniques. Furthermore, the traditional techniques mentioned above are often limited to empirical study and lack theoretical guarantees. Instead, this work designs a principled algorithm for deferral that effectively accounts for expert imbalance while preserving theoretical soundness and guarantees.

## B. Imbalance in Two-Stage Learning to Defer

**Notion of imbalance.** To define the notion of imbalance in the two-stage learning to defer setting, we refer readers to the definition of the deferral loss function $L_{\text{def}}$ (Section 3). In this context, $x$ denotes the input instance, and *imbalance* refers to a situation where, for a large majority of instances, the same expert consistently incurs the lowest cost—that is, is considered the most suitable expert according to the cost function used in minimizing the deferral loss. This results in a highly skewed deferral pattern that over-relies on a small subset of experts, potentially not effectively using others who may be better suited for specific parts of the input space.

The term *distribution of experts* refers to the empirical distribution over expert selections induced by the deferral policy across the input space. For example, in an LLM-based deferral system, imbalance may occur when one pretrained LLM performs significantly better than others across most inputs. If the deferral mechanism does not account for this, it may default to that model universally, ignoring specialized models that perform better on certain subpopulations.

Our experimental setup is designed to reflect precisely this kind of imbalance scenario—where a small subset of experts dominates unless appropriately regulated.

**Failure of standard deferral algorithms.** The observation that models trained on imbalanced datasets tend to underperform on minority categories is well-established in the imbalanced learning literature (Zhang et al., 2023). This phenomenon is often associated with *long-tailed* distributions, where a few dominant classes receive the majority of the training data, leading classifiers to perform poorly on underrepresented classes—sometimes only marginally outperforming naïve baselines that always predict the majority class.

In our context, this translates to a small subset of experts being favored across most of the input space, resulting in a similarly long-tailed distribution over expert selections. Consequently, deferral algorithms trained under such imbalance tend to overfit to the dominant experts and not effectively use others—mirroring patterns observed in general imbalanced learning.

While this issue has not been extensively studied in the specific context of two-stage learning to defer, our work builds on this well-known phenomenon to motivate the need for more balanced expert use. In particular, we present severe expert imbalance scenarios in Appendix C.3, where baseline methods fail, and show that our proposed algorithm, MILD, performs effectively to illustrate this point.

## C. Additional Experiments and Details

### C.1. Image Classification Setup Details

To systematically evaluate robustness to expert imbalance, we constructed three expert setups with varying degrees of coverage and overlap. These setups simulate scenarios where experts have specialized sub-domains of competence. In particular, experts are designed to exhibit varying performance across different labels, a specific form of imbalance. We adopt this setup because label-based separation provides a clear and interpretable way to model expert specialization, and it can also serve as a proxy for more general forms of input partitioning. Our methods remain effective in scenarios where expert performance varies according to other characteristics of the input space (e.g., image features rather than labels).

**Setup I: "7 vs. 2 vs. 1".** Three synthetic experts were available. Expert 1 was always correct for the first 70% of classes (e.g., classes 0–6 for CIFAR-10) and predicted uniformly at random for the remaining classes. Expert 2 was always correct for the next 20% of classes (e.g., classes 7–8 for CIFAR-10) and predicted randomly for the remaining classes. Expert 3 was

always correct for the last 10% of classes (e.g., class 9 for CIFAR-10) and predicted randomly for all other classes.

**Setup II: "5 vs. 2 vs. 2 vs. 1".** Four experts were available. Expert 1 was always correct for the first 50% of classes (e.g., classes 0–4 for CIFAR-10) and predicted uniformly at random for the remaining classes. Expert 2 was always correct for the next 20% of classes (e.g., classes 5–6 for CIFAR-10) and generated random predictions for the remaining classes. Expert 3 was always correct for the next 20% of classes (e.g., classes 7–8 for CIFAR-10) and generated random predictions for the remaining classes. Expert 4 was always correct for the last 10% of classes (e.g., class 9 for CIFAR-10) and predicted randomly for all other classes.

**Setup III: "4 vs. 2 vs. 2 vs. 1 vs. 1".** Expert 1 was always correct for the first 40% of classes and predicted randomly for the remaining classes. Expert 2 was always correct for the next 20% of classes and predicted randomly for the remaining classes. Expert 3 was always correct for the next 20% of classes and predicted randomly for the remaining classes. Expert 4 was always correct for the next 10% of classes and predicted randomly for all other classes. Expert 5 was always correct for the last 10% of classes and predicted randomly for all other classes.

We carried out experiments with two types of experts: synthetic and real. The synthetic experts were generated precisely as described in the three steps. Real experts, on the other hand, were trained on the training data from their corresponding classes plus a 1% fraction of training data from the other classes. For example, in the "7 vs. 2 vs. 1" setup on CIFAR-10, Expert 1 was trained using all training data from classes 0–6 and 1% of training data from classes 7–9.

We considered two types of cost functions. For the first type, we chose the misclassification errors of the experts as the cost functions: $c_j(x,y) = 1_{g_j(x) \neq y}$. For the second type, we chose the misclassification errors of the experts plus the percentage of the domain where the expert is accurate as the cost function. For example, in setup I ("7 vs. 2 vs. 1"), the cost functions are chosen as $c_1(x,y) = 1_{g_1(x) \neq y} + 0.7$, $c_2(x,y) = 1_{g_2(x) \neq y} + 0.2$ and $c_3(x,y) = 1_{g_3(x) \neq y} + 0.1$ for Expert 1, Expert 2, and Expert 3, respectively.

## C.2. LLM Routing on MMLU: Extended Details

We provide further details on the MMLU experimental setup used in Section 6.

**Models.** We adopted the `Qwen 2.5-Instruct` family of models (Yang et al., 2025) as experts due to their state-of-the-art performance in the open-weights category and their consistent architecture across sizes.

- **Expert 1 (Strong):** `Qwen2.5-7B-Instruct`. Used as the generalist anchor.

- **Expert 2 (Medium):** `Qwen2.5-1.5B-Instruct`. Represents a balanced edge-device model.

- **Expert 3 (Tiny):** `Qwen2.5-0.5B-Instruct`. Represents an extremely lightweight speculative model.

**Dataset.** We constructed the routing dataset using the **MMLU** (Massive Multitask Language Understanding) benchmark (Hendrycks et al., 2021). Specifically, we aggregated the 'high_school_mathematics' and 'high_school_world_history' subsets to create a mix of reasoning-heavy and knowledge-heavy queries. The test set consisted of 2,000 samples.

**Cost Settings.** We defined two distinct regimes to test the router's adaptability:

1. **Setting (a) Error Only:** The cost is binary, $c_k(x,y) = 1_{g_k(x) \neq y}$. This tests the router's ability to maximize accuracy without regard for compute. Since the 7B model is generally superior, this setting creates a massive imbalance where one expert is optimal $> 80\%$ of the time.

2. **Setting (b) Error + Cost:** The cost includes a normalized inference penalty, $c_k(x,y) = 1_{g_k(x) \neq y} + \beta_k$. We set $\beta = [1.0, 0.6, 0.1]$. This creates a complex trade-off: the 0.5B model is "optimal" (lowest loss) for any query it gets right, while the 7B model is only optimal for queries where both smaller models fail.

## C.3. Severe Expert Imbalance Analysis

To further stress-test the algorithms, we examine a scenario with severe expert imbalance, where the baseline method TDEF fails, while our proposed algorithm, MILD, performs effectively.

**Severe expert imbalance scenario:** For this scenario, we consider two experts, defining their cost functions based on misclassification errors: $c_j(x,y) = 1_{g_j(x) \neq y}$. This expert configuration, which reflects the real experts setting discussed in

*Table 4.* Comparison of our MILD algorithm with TDEF on CIFAR-10, CIFAR-100, SVHN under severe expert imbalance.

| Method | Dataset | Deferral Loss DL ~ error rate | Ratio of Expert Deferral (%) 1 | 2 |
|---|---|---|---|---|
| TDEF | CIFAR-10 | $0.0810 \pm 0.0025$ | 97.94 | 2.06 |
| MILD | | $\mathbf{0.0125 \pm 0.0016}$ | 90.52 | 9.48 |
| TDEF | CIFAR-100 | $0.0881 \pm 0.0034$ | 98.41 | 1.59 |
| MILD | | $\mathbf{0.0192 \pm 0.0037}$ | 91.56 | 8.44 |
| TDEF | SVHN | $0.0782 \pm 0.0016$ | 97.55 | 2.45 |
| MILD | | $\mathbf{0.0086 \pm 0.0012}$ | 90.05 | 9.95 |

Section 6, involved training Expert 1 with all training data from 90% of classes and 1% from the remaining classes. Expert 2 was similarly trained using all data from 10% of classes and 1% from the remaining classes.

Table 4 shows that a naive application of the TDEF algorithm in this scenario predominantly selects Expert 1, thereby failing to leverage Expert 2 despite its relevance to a nontrivial portion of the data. In contrast, MILD consistently achieves significantly smaller deferral losses by selecting the experts much closer to their optimal allocation.

### C.4. Choice of $\rho$

As discussed in Section 4.2 (B. General cost-sensitive algorithm), while our general algorithm allows $\rho_j$ to be tuned freely over a range of values, the search is in fact guided by the theoretically optimal values derived from our analysis. Specifically, the theoretical guidance suggests choosing $\rho_j$ close to $\frac{(m_j \mathsf{X}_j^2)^{\frac{1}{3}}}{\overline{\mathsf{X}}} \overline{\rho}$. In the experiments, we use this expression to initialize the search range, and perform validation-based tuning in a small neighborhood around these theoretically motivated values, using a step size of 1 over the interval $\left[ \frac{(m_j \mathsf{X}_j^2)^{\frac{1}{3}}}{\overline{\mathsf{X}}} - 5, \frac{(m_j \mathsf{X}_j^2)^{\frac{1}{3}}}{\overline{\mathsf{X}}} + 5 \right]$. Empirically, we observe that performance is quite robust within this neighborhood, which suggests that our theoretical estimation serves as a reliable prior for guiding the selection of $\rho_j$.

## D. Discussion on Alternative Baselines

We briefly address the applicability of confidence-based methods to the two-stage multiple-expert deferral setting. (For a discussion on standard cost-sensitive methods, see Section 6.2.)

**Confidence-based methods.** Standard baselines that rely on thresholding prediction confidence are primarily designed for the single-expert (or abstention) setting, where the decision is binary and costs are typically constant. Extending this paradigm to the multi-expert setting is non-trivial, as it is unclear how to define optimal thresholds when experts possess heterogeneous costs and predictive capabilities. To the best of our knowledge, the algorithm proposed by Mao et al. (2023a) is the only existing baseline specifically formulated for two-stage deferral with multiple experts.

While one might consider a *cascading* approach, where experts are queried sequentially based on confidence, this strategy is inherently sensitive to the ordering of experts and implicitly assumes a fixed ranking of predictive strength. Furthermore, it ignores the *instance-dependent* inference costs associated with each expert. In contrast, our method operates in a *routing* framework, selecting the most suitable expert from a parallel ensemble based on both accuracy and cost, regardless of expert ordering. Moreover, even in the simpler abstention setting, confidence-based methods have been shown to be suboptimal (Cortes et al., 2016a) and are outperformed by learning-to-defer approaches (Mao et al., 2023a). Crucially, neither approach accounts for expert imbalance, the central challenge addressed by our work.

# E. Reformulation of the Deferral Loss: Proofs of Lemma 3.1 and Lemma 3.2

**Lemma 3.1.** *For any $f \in \mathcal{F}$ and $(x, y) \in \mathcal{X} \times \mathcal{Y}$, the loss function $\mathsf{L}_{\mathrm{def}}$ can be expressed as follows:*

$$\mathsf{L}_{\mathrm{def}}(f, x, y) = \sum_{k=1}^{p} \left( \sum_{k'=1}^{p} c_{k'}(x, y) 1_{k' \neq k} \right) 1_{\rho_f(x,k) \leq 0} - (p-2) \sum_{k=1}^{p} c_k(x, y). \tag{1}$$

*Proof.* For any $f \in \mathcal{F}$ and $(x, y) \in \mathcal{X} \times \mathcal{Y}$, we have

$$
\begin{aligned}
\mathsf{L}_{\mathrm{def}}(f, x, y) &= \sum_{k=1}^{p} c_k(x, y) 1_{\mathsf{f}(x)=k} \\
&= \sum_{k=1}^{p} c_k(x, y) \left( \sum_{k'=1}^{p} 1_{\mathsf{f}(x) \neq k'} 1_{k' \neq k} - (p-2) \right) \\
&= \sum_{k=1}^{p} \sum_{k'=1}^{p} c_k(x, y) 1_{\mathsf{f}(x) \neq k'} 1_{k' \neq k} - (p-2) \sum_{k=1}^{p} c_k(x, y) \\
&= \sum_{k=1}^{p} \left( \sum_{k'=1}^{p} c_{k'}(x, y) 1_{k' \neq k} \right) 1_{\mathsf{f}(x) \neq k} - (p-2) \sum_{k=1}^{p} c_k(x, y) \\
&= \sum_{k=1}^{p} \left( \sum_{k'=1}^{p} c_{k'}(x, y) 1_{k' \neq k} \right) 1_{\rho_f(x,k) \leq 0} - (p-2) \sum_{k=1}^{p} c_k(x, y),
\end{aligned}
\tag{8}
$$

which completes the proof. $\qquad \square$

**Lemma 3.2.** *For any $f \in \mathcal{F}$ and $(x, y) \in \mathcal{X} \times \mathcal{Y}$, the loss function $\mathsf{L}_{\mathrm{def}}$ can be expressed as follows:*

$$\mathsf{L}_{\mathrm{def}}(f, x, y) = \sum_{k=1}^{p} (1 - c_k(x, y)) 1_{\rho_f(x,k) \leq 0} + \sum_{k=1}^{p} c_k(x, y) - (p-1). \tag{1}$$

*Proof.* For any $f \in \mathcal{F}$ and $(x, y) \in \mathcal{X} \times \mathcal{Y}$, we have

$$
\begin{aligned}
\mathsf{L}_{\mathrm{def}}(f, x, y) &= \sum_{k=1}^{p} c_k(x, y) 1_{\mathsf{f}(x)=k} \\
&= \sum_{k=1}^{p} c_k(x, y) \left( 1 - 1_{\mathsf{f}(x) \neq k} \right) \\
&= \sum_{k=1}^{p} c_k(x, y) - \sum_{k=1}^{p} (c_k(x, y) - 1) 1_{\mathsf{f}(x) \neq k} - \sum_{k=1}^{p} 1_{\mathsf{f}(x) \neq k} \\
&= \sum_{k=1}^{p} (1 - c_k(x, y)) 1_{\mathsf{f}(x) \neq k} + \sum_{k=1}^{p} c_k(x, y) - (p-1) \\
&= \sum_{k=1}^{p} (1 - c_k(x, y)) 1_{\rho_f(x,k) \leq 0} + \sum_{k=1}^{p} c_k(x, y) - (p-1),
\end{aligned}
\tag{9}
$$

which completes the proof. $\qquad \square$

# F. Margin Bound: Proof of Theorem 4.1

**Theorem 4.1** (Margin bound for imbalanced cost-sensitive classification). *Let $\mathcal{F}$ be a family of functions mapping from $\mathcal{X} \times [p]$ to $\mathbb{R}$, and fix $\boldsymbol{\rho} = [\rho_k]_{k \in [p]}$. Then, for any $\delta > 0$, with probability at least $1 - \delta$, each of the following inequalities holds for all $f \in \mathcal{F}$:*

$$\mathcal{E}_{\mathsf{L}}(f) \le \widehat{\mathcal{E}}_{S,\boldsymbol{\rho}}(f) + 4\sqrt{2p}\,\mathfrak{R}_{m,\boldsymbol{\rho}}(\mathcal{F}) + \sqrt{\frac{\log \frac{1}{\delta}}{2m}}$$

$$\mathcal{E}_{\mathsf{L}}(f) \le \widehat{\mathcal{E}}_{S,\boldsymbol{\rho}}(f) + 4\sqrt{2p}\,\widehat{\mathfrak{R}}_{S,\boldsymbol{\rho}}(\mathcal{F}) + 3\sqrt{\frac{\log \frac{2}{\delta}}{2m}}.$$

*Proof.* Consider the family of functions taking values in $[0, 1]$:

$$\mathcal{F}' = \{z = (x, k) \mapsto \mathsf{L}_{\boldsymbol{\rho}}(f, x, k) \colon f \in \mathcal{F}\}.$$

By (Mohri et al., 2018, Theorem 3.3), with probability at least $1 - \delta$, for all $g \in \mathcal{F}'$,

$$\mathbb{E}[g(z)] \le \frac{1}{m}\sum_{i=1}^{m} g(z_i) + 2\widehat{\mathfrak{R}}_S(\mathcal{F}') + 3\sqrt{\frac{\log \frac{2}{\delta}}{2m}},$$

and thus, for all $f \in \mathcal{F}$,

$$\mathbb{E}[\mathsf{L}_{\boldsymbol{\rho}}(f, x, k)] \le \widehat{\mathcal{E}}_{S,\boldsymbol{\rho}}(f) + 2\widehat{\mathfrak{R}}_S(\mathcal{F}') + 3\sqrt{\frac{\log \frac{2}{\delta}}{2m}}.$$

Since $\mathcal{E}_{\mathsf{L}}(f) \le \mathcal{E}_{\mathsf{L}_{\boldsymbol{\rho}}}(f) = \mathbb{E}[\mathsf{L}_{\boldsymbol{\rho}}(f, x, k)]$, we have

$$\mathcal{E}_{\mathsf{L}}(f) \le \widehat{\mathcal{E}}_{S,\boldsymbol{\rho}}(f) + 2\widehat{\mathfrak{R}}_S(\mathcal{F}') + 3\sqrt{\frac{\log \frac{2}{\delta}}{2m}}.$$

Fix $f$, $(x_i, k_i)$ and $\rho > 0$, define $\Psi$ as follows:

$$\Psi\big([f(x_i, k)]_{k \in [p]}\big) = \max_{k' \in [p]} \{c(x_i, k_i, k')\Phi_\rho(f(x_i, k_i) - f(x_i, k'))\}.$$

Then, by the sub-additivity of the maximum operator, we can write for any $f, \widetilde{f} \in \mathcal{F}$:

$$\Psi\big([f(x_i, k)]_{k \in [p]}\big) - \Psi\big([\widetilde{f}(x_i, k)]_{k \in [p]}\big)$$
$$\le \max_{k' \in [p]} \big\{c(x_i, k_i, k')\Phi_\rho(f(x_i, k_i) - f(x_i, k')) - c(x_i, k_i, k')\Phi_\rho\big(\widetilde{f}(x_i, k_i) - \widetilde{f}(x_i, k')\big)\big\}$$
$$\le \max_{k' \in [p]} \left\{\frac{2c(x_i, k_i, k')}{\rho}\big\|[f(x_i, k) - \widetilde{f}(x_i, k)]_{k \in [p]}\big\|_1\right\} \qquad \text{(by $\frac{1}{\rho}$-Lipschitzness of $\Phi_\rho$)}$$
$$\le \frac{2\sqrt{p}}{\rho}\big\|[f(x_i, k) - \widetilde{f}(x_i, k)]_{k \in [p]}\big\|_2.$$

Thus, $\Psi$ is $\frac{2\sqrt{p}}{\rho}$-Lipschitz with respect to the $\|\cdot\|_2$ norm. Thus, by the vector contraction lemma (Maurer, 2016; Cortes et al., 2016c), $\widehat{\mathfrak{R}}_S(\mathcal{F}')$ can be bounded as follows:

$$\widehat{\mathfrak{R}}_S(\mathcal{F}') \le \frac{2\sqrt{2p}}{m}\,\mathbb{E}_{\boldsymbol{\epsilon}}\left[\sup_{f \in \mathcal{F}}\left\{\sum_{j=1}^{p}\sum_{i \in S_j}\sum_{k=1}^{p} \epsilon_{ik}\frac{f(x_i, k)}{\rho_j}\right\}\right] = 2\sqrt{2p}\,\widehat{\mathfrak{R}}_{S,\boldsymbol{\rho}}(\mathcal{F}).$$

This proves the second inequality. The first inequality, can be derived in the same way by using the first inequality of (Mohri et al., 2018, Theorem 3.3). $\qquad\square$

# G. Derivation of Surrogate Losses

Using the fact that $c(x, k, k') = 0$ for $k = k'$, we can rewrite $\mathsf{L}_\rho$ as

$$\mathsf{L}_\rho(f, x, k) = \max_{k' \neq k}\{c(x, k, k')\Phi_{\rho_k}(f(x, k) - f(x, k'))\}.$$

For $\Phi_\rho(t) \leq \Psi(t/\rho)$, we have:

$$\mathsf{L}_\rho(f, x, k) \leq \max_{k' \neq k}\left\{c(x, k, k')\Psi\left(\frac{f(x, k) - f(x, k')}{\rho_k}\right)\right\}.$$

In particular, choosing $\Psi$ to be the logistic loss yields:

$$\mathsf{L}_\rho(f, x, k) \leq \max_{k' \neq k}\left\{c(x, k, k')\log\left[1 + \exp\left(\frac{f(x, k') - f(x, k)}{\rho_k}\right)\right]\right\}.$$

When $c(x, k, k')$ is independent of $(k, k')$, we can write:

$$\mathsf{L}_\rho(f, x, k) \leq \max_{k' \neq k}\left\{C(x)\log\left[1 + \exp\left(\frac{f(x, k') - f(x, k)}{\rho_k}\right)\right]\right\}$$

$$= C(x)\max_{k' \neq k}\left\{\log\left[1 + \exp\left(\frac{f(x, k') - f(x, k)}{\rho_k}\right)\right]\right\}$$

$$= C(x)\log\left[1 + \max_{k' \neq k}\exp\left(\frac{f(x, k') - f(x, k)}{\rho_k}\right)\right]$$

$$\leq C(x)\log\left[1 + \sum_{k' \neq k}\exp\left(\frac{f(x, k') - f(x, k)}{\rho_k}\right)\right]$$

$$= C(x)\log\left[\sum_{k'=1}^{p}\exp\left(\frac{f(x, k') - f(x, k)}{\rho_k}\right)\right].$$

Otherwise,

$$\mathsf{L}_\rho(f, x, k) \leq \max_{k' \neq k}\left\{c(x, k, k')\log\left[1 + \exp\left(\frac{f(x, k') - f(x, k)}{\rho_k}\right)\right]\right\}$$

$$= \log\left[\max_{k' \neq k}\left[1 + \exp\left(\frac{f(x, k') - f(x, k)}{\rho_k}\right)\right]^{c(x, k, k')}\right]$$

$$\leq \log\left[\sum_{k' \neq k}\left[1 + \exp\left(\frac{f(x, k') - f(x, k)}{\rho_k}\right)\right]^{c(x, k, k')}\right].$$

Another choice is to set $\Psi$ as the comp-sum loss with $\tau \neq 1$, which yields:

$$\mathsf{L}_\rho(f, x, k) \leq \max_{k' \neq k}\left\{c(x, k, k')\Phi^\tau\left[\exp\left(\frac{f(x, k') - f(x, k)}{\rho_k}\right)\right]\right\}$$

$$= \frac{1}{1 - \tau}\max_{k' \neq k}\left\{c(x, k, k')\left(\left[1 + \exp\left(\frac{f(x, k') - f(x, k)}{\rho_k}\right)\right]^{1-\tau} - 1\right)\right\}$$

$$\leq \frac{1}{1 - \tau}\sum_{k' \neq k}\left\{c(x, k, k')\left(\left[1 + \exp\left(\frac{f(x, k') - f(x, k)}{\rho_k}\right)\right]^{1-\tau} - 1\right)\right\}.$$

When $c(x, k, k')$ is independent of $(k, k')$, we can write:

$$\mathsf{L}_\rho(f, x, k) \leq \frac{C(x)}{1 - \tau}\sum_{k' \neq k}\left\{\left(\left[1 + \exp\left(\frac{f(x, k') - f(x, k)}{\rho_k}\right)\right]^{1-\tau} - 1\right)\right\}.$$

# H. $\mathcal{F}$-Consistency Bound: Proof of Theorem 5.1

**Theorem 5.1** ($\mathcal{F}$-consistency bound for imbalanced cost-sensitive surrogate loss)**.** *Let $\mathcal{F}$ be a complete hypothesis set. Then, for all $f \in \mathcal{F}$, $\rho > 0$, the following inequality holds:*

$$\mathcal{E}_{\mathsf{L}}(f) - \mathcal{E}_{\mathsf{L}}^*(\mathcal{F}) + \mathcal{M}_{\mathsf{L}}(\mathcal{F}) \le \sqrt{2}\Big(\mathcal{E}_{\widetilde{\mathsf{L}}_\rho}(f) - \mathcal{E}_{\widetilde{\mathsf{L}}_\rho}^*(\mathcal{F}) + \mathcal{M}_{\widetilde{\mathsf{L}}_\rho}(\mathcal{F})\Big)^{\frac{1}{2}}.$$

*Proof.* The conditional error and the best-in-class conditional error of the cost-sensitive zero-one loss can be expressed as follows:

$$\mathbb{E}_k[\mathsf{L}(f,x,k) \mid x] = C(x) \sum_{k \in [p]} \mathsf{p}(k|x) 1_{\rho_f(x,y) \le 0} = C(x)(1 - \mathsf{p}(\mathsf{f}(x)|x)),$$

$$\inf_{f \in \mathcal{F}} \mathbb{E}_k[\mathsf{L}(f,x,k) \mid x] = C(x)\Big(1 - \max_{k \in [p]} \mathsf{p}(k|x)\Big).$$

Let $k_{\max} = \mathrm{argmax}_{k \in [p]} \mathsf{p}(k|x)$. Then, the difference between the two terms is given by:

$$\mathbb{E}_k[\mathsf{L}(f,x,k) \mid x] - \inf_{f \in \mathcal{F}} \mathbb{E}_k[\mathsf{L}(f,x,k) \mid x] = \mathsf{p}(k_{\max}|x) - \mathsf{p}(\mathsf{f}(x)|x).$$

For the imbalanced cost-sensitive surrogate, the conditional error can be expressed as follows:

$$\mathbb{E}_k\big[\widetilde{\mathsf{L}}_\rho(f,x,k) \mid x\big]$$
$$= \sum_{k \in [p]} \mathsf{p}(k|x)\Phi_{\rho_y}(\rho_f(x,y))$$
$$= C(x) \sum_{k \in [p]} \mathsf{p}(k|x)\bigg\{\log\bigg[\sum_{k'=1}^{p} \exp\bigg(\frac{f(x,k') - f(x,k)}{\rho_k}\bigg)\bigg]\bigg\}$$
$$= C(x)\mathsf{p}(k_{\max}|x) \log\bigg(\sum_{k' \in [p]} e^{\frac{f(x,k') - f(x,k_{\max})}{\rho_{k_{\max}}}}\bigg) + C(x)\mathsf{p}(\mathsf{f}(x)|x) \log\bigg(\sum_{k' \in [p]} e^{\frac{f(x,k') - f(x,\mathsf{f}(x))}{\rho_{\mathsf{f}(x)}}}\bigg)$$
$$+ C(x) \sum_{k \notin \{k_{\max}, \mathsf{f}(x)\}} \mathsf{p}(k|x) \log\bigg(\sum_{k' \in [p]} e^{\frac{f(x,k') - f(x,k)}{\rho_k}}\bigg).$$

For any $f \in \mathcal{F}$ and $x \in \mathcal{X}$, by the completeness of $\mathcal{F}$, we can always find a family of hypotheses $\{\overline{f}_\mu : \mu \in \mathbb{R}\} \subset \mathcal{F}$ such that $\overline{f}_\mu(x, \cdot)$ take the following values:

$$\overline{f}_\mu(x,k) = \begin{cases} f(x,k) & \text{if } k \notin \{k_{\max}, \mathsf{f}(x)\} \\ \log(\exp[f(x,k_{\max})] + \mu) & \text{if } k = \mathsf{f}(x) \\ \log(\exp[f(x,\mathsf{f}(x))] - \mu) & \text{if } k = k_{\max}. \end{cases} \tag{10}$$

Note that the hypotheses $\overline{f}_\mu$ has the following property:

$$\sum_{k \in [p]} e^{\frac{f(x,k)}{\rho_{k'}}} = \sum_{k \in [p]} e^{\frac{\overline{f}_\mu(x,k)}{\rho_{k'}}}, \ \forall \mu \in \mathbb{R} \text{ and } k' \in [p]. \tag{11}$$

Thus, the best-in-class conditional error can be upper bounded as follows:

$$\inf_{f \in \mathcal{F}} \mathbb{E}_k\big[\widetilde{\mathsf{L}}_\rho(f,x,k) \mid x\big] \le \inf_{\mu \in \mathbb{R}} \mathbb{E}_k\big[\widetilde{\mathsf{L}}_\rho(\overline{f}_\mu,x,k) \mid x\big].$$

The conditional regret can be lower bounded as follows:

$$\mathbb{E}_k\Big[\widetilde{\mathsf{L}}_{\boldsymbol{\rho}}(f,x,k)\mid x\Big] - \inf_{f\in\mathcal{F}}\mathbb{E}_k\Big[\widetilde{\mathsf{L}}_{\boldsymbol{\rho}}(f,x,k)\mid x\Big]$$

$$\geq \mathbb{E}_k\Big[\widetilde{\mathsf{L}}_{\boldsymbol{\rho}}(f,x,k)\mid x\Big] - \inf_{\mu\in\mathbb{R}}\mathbb{E}_k\Big[\widetilde{\mathsf{L}}_{\boldsymbol{\rho}}(\overline{f}_\mu,x,k)\mid x\Big]$$

$$= \sup_{\mu\in\mathbb{R}}\left\{\mathsf{p}(k_{\max}|x)\left(\log\left(\frac{\sum_{k'\in[p]}e^{\frac{f(x,k')}{\rho_{k_{\max}}}}}{e^{\frac{f(x,k_{\max})}{\rho_{k_{\max}}}}}\right) - \log\left(\frac{\sum_{k'\in[p]}e^{\frac{f(x,k')}{\rho_{k_{\max}}}}}{e^{\frac{f(x,\mathsf{f}(x))-\mu}{\rho_{k_{\max}}}}}\right)\right)\right.$$

$$\left. + \mathsf{p}(\mathsf{f}(x)|x)\left(\log\left(\frac{\sum_{k'\in[p]}e^{\frac{f(x,k')}{\rho_{\mathsf{f}(x)}}}}{e^{\frac{f(x,\mathsf{f}(x))}{\rho_{\mathsf{f}(x)}}}}\right) - \log\left(\frac{\sum_{k'\in[p]}e^{\frac{f(x,k')}{\rho_{\mathsf{f}(x)}}}}{e^{\frac{f(x,k_{\max})+\mu}{\rho_{\mathsf{f}(x)}}}}\right)\right)\right\}$$

$$\geq \mathsf{p}(k_{\max}|x)\log\left[\frac{\big(e^{f(x,k_{\max})}+e^{f(x,\mathsf{f}(x))}\big)\mathsf{p}(k_{\max}|x)}{e^{f(x,k_{\max})}\big(\mathsf{p}(k_{\max}|x)+\mathsf{p}(\mathsf{f}(x)|x)\big)}\right]$$

$$ + \mathsf{p}(\mathsf{f}(x)|x)\log\left[\frac{\big(e^{f(x,k_{\max})}+e^{f(x,\mathsf{f}(x))}\big)\mathsf{p}(\mathsf{f}(x)|x)}{e^{f(x,\mathsf{f}(x))}\big(\mathsf{p}(k_{\max}|x)+\mathsf{p}(\mathsf{f}(x)|x)\big)}\right]$$

(Maximum is achieved by $\mu^* = \frac{e^{f(x,\mathsf{f}(x))}\mathsf{p}(\mathsf{f}(x)|x)\rho_{\mathsf{f}(x)} - e^{f(x,k_{\max})}\mathsf{p}(k_{\max}|x)\rho_{k_{\max}}}{\mathsf{p}(k_{\max}|x)\rho_{k_{\max}}+\mathsf{p}(\mathsf{f}(x)|x)\rho_{\mathsf{f}(x)}}$)

$$\geq \mathsf{p}(k_{\max}|x)\log\left[\frac{2\mathsf{p}(k_{\max}|x)}{\big(\mathsf{p}(k_{\max}|x)+\mathsf{p}(\mathsf{f}(x)|x)\big)}\right] + \mathsf{p}(\mathsf{f}(x)|x)\log\left[\frac{2\mathsf{p}(\mathsf{f}(x)|x)}{\big(\mathsf{p}(k_{\max}|x)+\mathsf{p}(\mathsf{f}(x)|x)\big)}\right]$$

(minimum is attained when $f(x,\mathsf{f}(x)) = f(x,k_{\max})$)

$$\geq \big[\mathsf{p}(k_{\max}|x)+\mathsf{p}(\mathsf{f}(x)|x)\big] \times \frac{1}{2}\left[\left|\frac{\mathsf{p}(k_{\max}|x)}{\mathsf{p}(k_{\max}|x)+\mathsf{p}(\mathsf{f}(x)|x)} - \frac{1}{2}\right| + \left|\frac{\mathsf{p}(\mathsf{f}(x)|x)}{\mathsf{p}(k_{\max}|x)+\mathsf{p}(\mathsf{f}(x)|x)} - \frac{1}{2}\right|\right]^2$$

(Pinsker's inequality (Mohri et al., 2018, Proposition E.7))

$$= \big[\mathsf{p}(k_{\max}|x)+\mathsf{p}(\mathsf{f}(x)|x)\big] \times \frac{1}{2}\left[\frac{\mathsf{p}(k_{\max}|x)-\mathsf{p}(\mathsf{f}(x)|x)}{\mathsf{p}(k_{\max}|x)+\mathsf{p}(\mathsf{f}(x)|x)}\right]^2 \qquad (\mathsf{p}(k_{\max}|x)\geq\mathsf{p}(\mathsf{f}(x)|x))$$

$$\geq \frac{1}{2}\big(\mathsf{p}(k_{\max}|x)-\mathsf{p}(\mathsf{f}(x)|x)\big)^2. \qquad (\mathsf{p}(k_{\max}|x)+\mathsf{p}(\mathsf{f}(x)|x)\leq 1)$$

By taking the expectation of both sides and using Jensen's inequality, we obtain:

$$\mathcal{E}_{\mathsf{L}}(f) - \mathcal{E}_{\mathsf{L}}^*(\mathcal{F}) + \mathcal{M}_{\mathsf{L}}(\mathcal{F}) \leq \sqrt{2}\Big(\mathcal{E}_{\widetilde{\mathsf{L}}_\rho}(f) - \mathcal{E}_{\widetilde{\mathsf{L}}_\rho}^*(\mathcal{F}) + \mathcal{M}_{\widetilde{\mathsf{L}}_\rho}(\mathcal{F})\Big)^{\frac{1}{2}},$$

which completes the proof. $\qquad\qquad\square$

# I. Cost-Sensitive Learning and Structured Prediction

While our primary focus is addressing class imbalance in learning to defer, our theoretical analysis and algorithm offer independent contributions to cost-sensitive learning and related areas such as structured prediction. We briefly discuss that in this section.

We introduced a margin-based upper bound on the cost-sensitive loss function:

$$\mathsf{L}(f,x,k) \leq \max_{k'\in[p]}\Big\{c(x,k,k')\Phi_\rho\big(f(x,k)-f(x,k')\big)\Big\}.$$

This bound extends prior work by incorporating instance-dependent cost functions ($x$-dependent), making it both more general and tighter than the upper bound used in (Cortes, Kuznetsov, Mohri, and Yang, 2016c) (see Lemma 4 and the surrogate losses on page 8), the closest related study on margin-based cost-sensitive and structured prediction bounds.

For instance, adopting the hinge loss for $\Psi$ as an auxiliary margin-based loss to upper bound $\mathsf{L}_\rho$ yields a loss function that serves as a lower bound for the hinge-loss-type surrogate loss in that publication (which coincides with the StructSVM loss function), even in the balanced case ($\rho_k = \rho$ for all $k$). Similarly, using the logistic loss for $\Psi$ as an auxiliary margin-based

loss function to upper bound $L_\rho$ results in a loss function that is upper bounded by the logistic-loss-type surrogate in (Cortes, Kuznetsov, Mohri, and Yang, 2016c) (an extension of Conditional Random Field (CRF) loss function), even in the balanced case.

For the same reasons, our margin-based theoretical analysis yields more favorable learning bounds, since our margin loss $L_\rho$ serves as a lower bound for the multiplicative margin loss considered in (Cortes, Kuznetsov, Mohri, and Yang, 2016c), even in the balanced case. Thus, this leads to improved learning bounds for structured prediction compared to that study, as well as structured prediction algorithms with stronger theoretical guarantees.

We leave a more detailed study of the theoretical and algorithmic implications of our analysis for cost-sensitive learning and structured prediction to future work.

