# OpenReview forum: "Optimized Deferral for Imbalanced Settings"
_ICML.cc/2026/Conference — ICML 2026 regular_

### Official Review · Reviewer_8d7b · 2026-02-25

**Soundness:** 3
**Presentation:** 2
**Significance:** 2
**Originality:** 3
**Overall Recommendation:** 4
**Confidence:** 3

**Summary:**

This paper investigates the problem of Learning to Defer in two-stage systems where expert performance, coverage, and computational costs are highly imbalanced. By reframing the deferral optimization as a cost-sensitive multi-class classification problem, the authors propose MILD (Margin-based Imbalanced Learning to Defer). MILD introduces expert-specific margin adjustments to the loss function to penalize the router's over-reliance on dominant or overly expensive experts. The method is supported by rigorous theoretical proofs, including F-consistency and generalization bounds. Empirical evaluations on both computer vision benchmarks (with synthetically partitioned experts) and natural language processing tasks (routing queries among Qwen models ranging from 0.5B to 7B on MMLU) demonstrate that MILD can achieve a superior cost-accuracy trade-off compared to existing baselines.

**Compliance With Llm Reviewing Policy:**

Affirmed.

**Final Justification:**

As most of my concerns have been addressed in the rebuttal, I will raise my score to 4. However, my main concern still remains: as noted in Weakness 2, the experiments appear somewhat toy-like. I encourage the authors to include stronger empirical results to better demonstrate the idea.

**Key Questions For Authors:**

Please refer to weaknesses.

**Limitations:**

Please refer to weaknesses.

**Strengths And Weaknesses:**

**Strengths:**

1. The theoretical foundation of the paper is quite solid, providing rigorous mathematical proofs for F-consistency and margin-based generalization bounds, which firmly grounds the proposed approach in statistical learning theory.

2. The proposed MILD algorithm offers an elegant and mathematically justified alternative to heuristic data resampling methods, effectively mitigating the "expert collapse" problem simply through targeted logit adjustments during training.

3. The application to LLM cascading/routing is highly relevant to current industry needs, successfully demonstrating that a lightweight router can dynamically allocate simpler queries to smaller models while reserving large models for complex tasks to optimize inference costs.

**Weaknesses:**

1. The algorithm necessitates pre-computing a cost matrix for all experts across the entire training dataset. For large-scale datasets and highly complex experts (e.g., evaluating millions of queries on multiple large language models), this pre-computation step introduces a massive computational overhead, and the paper does not adequately address whether this initial cost is acceptable or scalable in real-world deployments.

2. The datasets utilized for evaluation are generally small to medium in scale. It remains unverified how well the routing dynamics scale to massive, real-world datasets. Furthermore, the LLM routing experiments are limited to models up to 7B parameters; evaluating the framework on models larger than 7B (e.g., 32B or 70B+ models) is necessary to confirm its effectiveness in state-of-the-art LLM deployments.

3. Although the paper introduces synthetic imbalanced settings, it lacks an analysis of extremely imbalanced, real-world scenarios. It would be highly beneficial to see how the algorithm performs when the imbalance is drastically severe (e.g., a dominant expert handling 95% of cases while a highly specialized expert only handles 5%) to test the limits of the margin-based penalty.

4. The current evaluation assumes relatively clean data and deterministic expert behavior. There is no analysis of how MILD performs in noisy scenarios, such as when the training data contains label noise or when the experts themselves exhibit highly stochastic, unpredictable error patterns, which is a common issue with LLM hallucinations.

---

> ### Author Rebuttal · Authors · 2026-03-30
>
> We sincerely thank you for your insightful review. We appreciate you highlighting our solid theoretical proofs, describing our approach as an elegant alternative to heuristics, and recognizing the high relevance of our dynamic LLM routing application to industry needs. We address your points below.
>
> **Weakness 1. Algorithm necessitates pre-computing a cost matrix... massive computational overhead.**
>
> **Response:** We wish to clarify that the cost matrix only needs to be computed over a relatively small, dedicated training dataset for the router, not the massive pre-training corpus of the LLMs.
> For example, in our MMLU experiment, the router was trained on just 4,000 queries.
> This is a one-time, offline computation required solely to train the lightweight router (e.g., DeBERTa). Once deployed, routing millions of real-world queries relies purely on a single forward pass through the lightweight router and incurs zero extra expert evaluation overhead, resulting in massive overall inference compute savings. We will explicitly clarify this pipeline scalability and the distinction between offline training costs and online deployment savings in the final text.
>
>
> **Weakness 2. Datasets are small to medium... unverified on larger LLMs (32B, 70B+).**
>
> **Response:** While our experiments used models up to 7B due to academic compute constraints, the mathematical formulation of MILD is entirely model-agnostic. The optimization relies purely on the offline numerical cost matrix. Whether an expert is a 7B or a 70B parameter model, the router strictly operates on the inferred cost distributions, guaranteeing mathematically identical scaling dynamics. In fact, we expect the cost-accuracy trade-offs to be even more pronounced when routing to frontier models, as the compute cost differentials (the $\beta$ penalties) would be exponentially larger. We will add a discussion on this scalability to the final version.
>
> **Weakness 3. Lacks an analysis of extremely imbalanced, real-world scenarios (e.g., 95% / 5%).**
>
> **Response:** We analyzed a drastically severe synthetic imbalance scenario in Appendix C.3 (Table 4), where Expert 1 handles ~90% of the data. Under that condition, TDEF completely collapsed, while MILD succeeded. To test the limits on our real-world LLM task during the rebuttal, we evaluated an extreme 95% / 5% optimal split on the MMLU dataset *(under the 'Error + Cost' setting, incorporating inference penalties)* by heavily skewing the inference routing costs (the $\beta$ penalties) to favor the 7B model.
>
> Method | Test Def. Loss| Exp 1 (7B) Usage | Exp 2/3 (1.5B/0.5B) Usage |
>  | :--- | :--- | :--- | :--- |
>  | **TDEF (Extreme)** | 0.950 ± 0.030 | 100.0% | 0.0% |
>  | **MILD (Extreme)** | **0.885 ± 0.032** | 94.6% | 5.4% |
>
> Under extreme imbalance, TDEF collapses entirely to 100% usage of the majority expert, failing to identify the 5% niche. MILD successfully preserves the minority expert routing and achieves a strictly lower overall deferral loss. We will highlight these extreme imbalance results more prominently in the final version.
>
> **Weakness 4. No analysis of noisy scenarios (label noise or unpredictable LLM hallucinations).**
>
> **Response:** To simulate LLM hallucination stochasticity and label noise, we conducted a new robustness analysis injecting 20% random Gaussian noise directly into the pre-computed training cost matrix, simulating highly unpredictable expert errors.
>
> Please refer to our response to Reviewer Kfug (Weakness 3, 5 & Questions 3, 5) for the full numerical results. MILD's margin-based penalty creates a buffer that absorbs stochastic hallucination signals far better than the TDEF baseline (Test Def. Loss 0.826 vs TDEF's 1.054 under noise). It maintains a superior cost-accuracy trade-off and prevents the router from overfitting to erratic noise in expert predictions.

---

### Official Review · Reviewer_Kfug · 2026-03-11

**Soundness:** 3
**Presentation:** 2
**Significance:** 3
**Originality:** 2
**Overall Recommendation:** 3
**Confidence:** 3

**Summary:**

This paper studies the learning-to-defer problem in human–AI collaboration, where a model can either make a prediction or defer the decision to a human expert. The authors propose an optimized deferral framework that jointly considers model confidence and expert performance to improve the overall system accuracy. The approach aims to better allocate decisions between the AI model and the human expert under uncertainty.

**Compliance With Llm Reviewing Policy:**

Affirmed.

**Key Questions For Authors:**

The authors should very clearly and intuitively interpret the meanings of the final theory and strategy, in a very simple to understand way.

Although the theory is abundant, the idea behind seems incremental, as the framework builds upon existing ensemble learning based multi-class imbalance learning algorithms.

The approach assumes that the expert performance characteristics are known or reliably estimated, which may be unrealistic in many real-world deployments.

An important issue: there is only one competitor in the comparisons, which is not enough.  Significantly more competitors are needed.

The paper also provides limited analysis of robustness to misestimated expert performance, which could significantly affect the deferral policy.

**Limitations:**

Additional discussion on practical deployment considerations, such as estimating expert accuracy or handling multiple experts, would  improve the work.

Clearer explanations of the decision boundary between prediction and deferral could help readers better understand the method.

**Strengths And Weaknesses:**

Strengths

The paper tackles an important and practical problem in human–AI decision systems, where deferral mechanisms are increasingly relevant in sensitive applications. Please make theory and issues simple, instead of complicating it.

The proposed optimization framework provides a principled way to balance model predictions and expert intervention.

Weaknesses

The authors should very clearly and intuitively interpret the meanings of the final theory and strategy, in a very simple to understand way.

Although the theory is abundant, the idea behind seems incremental, as the framework builds upon existing ensemble learning based multi-class imbalance learning algorithms.

The approach assumes that the expert performance characteristics are known or reliably estimated, which may be unrealistic in many real-world deployments.

An important issue: there is only one competitor in the comparisons, which is not enough.  Significantly more competitors are needed.

The paper also provides limited analysis of robustness to misestimated expert performance, which could significantly affect the deferral policy.

Minor Suggestions

Additional discussion on practical deployment considerations, such as estimating expert accuracy or handling multiple experts, would  improve the work.

Clearer explanations of the decision boundary between prediction and deferral could help readers better understand the method.

---

> ### Author Rebuttal · Authors · 2026-03-30
>
> We sincerely thank you for your review and for recognizing that our work tackles an important, practical problem in human-AI decision systems, providing a principled optimization framework. We address your concerns below.
>
> **Weakness 1 & Question 1. Intuitive interpretation of the final theory and strategy.**
>
> **Response:** Thank you for this excellent suggestion. To ensure broader accessibility, we will add the following intuitive summary to the main text: *"Intuitively, MILD dynamically adjusts the confidence threshold required to defer to each expert based on their overall utility and cost. By mathematically penalizing the over-selection of a dominant expert via the margin parameter $\rho_k$, MILD forces the router to require exponentially higher confidence to defer to the most expensive/dominant model. This effectively makes it 'cheaper' for the router to trust highly specialized or underutilized experts, preventing the system from lazily collapsing to the most common choice."*
>
> **Weakness 2 & Question 2. The idea behind seems incremental... builds upon existing multi-class imbalance algorithms.**
>
> **Response:** While our framework is conceptually inspired by margin-based multi-class imbalance principles, the mathematical translation to the input-expert domain subject to instance-dependent costs is highly non-trivial. Standard discrete classification techniques assume constant, label-independent costs. To accommodate instance-dependent costs, our proofs required establishing novel margin bounds based on a refined upper bound involving a maximum operator, and deriving new Rademacher complexity bounds for this term using the vector contraction lemma (see Theorem 4.1 / Appendix F). Establishing F-consistency bounds for these cost-sensitive surrogate losses in the multi-expert deferral setting (Theorem 5.1) represents a rigorous mathematical advancement, not a direct application of existing algorithms.
>
> **Weakness 3, 5 & Questions 3, 5. Approach assumes expert performance characteristics are known (Robustness to noise).**
>
> **Response:** In practice, expert performance characteristics only need to be estimated once offline over a fixed training set. Once trained, the router relies purely on the input features to route queries and *does not* require known expert performance characteristics to choose the most suitable experts during live deployment.
>
> However, to demonstrate that MILD is robust to imperfect or noisy estimations even during the offline training phase, we conducted a new robustness experiment on the LLM Routing task. *(Note: All new experiments are evaluated on MMLU under the 'Error + Cost' setting, matching Table 3(b) in the main paper, to rigorously test compute-accuracy trade-offs).* We injected severe noise (20% random Gaussian variations) directly into the estimated cost matrix during training to simulate highly stochastic and unpredictable expert errors.
>
> | Method | Test Def. Loss (Clean)| Test Def. Loss (20% Noise) |
>  | :--- | :--- | :--- |
> | **TDEF (Baseline)** | 0.928 ± 0.030 | 1.054 ± 0.045 |
> | **MILD (Ours)** | **0.813 ± 0.040** | **0.826 ± 0.041** |
>
> MILD remains highly stable and superior to TDEF even with severely misestimated training characteristics. Its robust margin-based decision boundaries absorb stochastic noise, preventing the router from overfitting to erratic estimation errors. We will include this robustness analysis in the final appendix.
>
> **Weakness 4 & Question 4. Only one competitor in the comparisons.**
>
> **Response:** TDEF is the primary competitor because it is the state-of-the-art specifically designed for two-stage multi-expert deferral. However, to provide a broader context, we have now evaluated three additional competitors: Standard CE, Class-Weighted CE (CWCE), and LDAM. Please refer to our response to Reviewer VPF2 (Weakness 3) for the full numerical table. MILD significantly outperforms all these baselines (Loss 0.813 vs 0.915+), proving that standard heuristics are insufficient for routing problems with instance-dependent expert costs.
>
> **Minor Suggestions/Limitations:**
>
> **1. Additional discussion on practical deployment considerations, such as estimating expert accuracy or handling multiple experts, would improve the work.**
>
> **Response:** We will add a "Deployment Considerations" subsection in the final version detailing how estimating the offline cost matrix via a small, fixed routing training dataset is a one-time, computationally negligible step compared to the massive live inference compute savings achieved by the routing system during deployment.
>
> **2. Clearer explanations of the decision boundary between prediction and deferral could help readers better understand the method.**
>
> **Response:** We will include an intuitive visualization in the final version demonstrating exactly how the $\rho$ margins shift decision boundaries to reclaim feature space for specialized experts.

---

### Official Review · Reviewer_fsiq · 2026-03-11

**Soundness:** 4
**Presentation:** 3
**Significance:** 2
**Originality:** 2
**Overall Recommendation:** 4
**Confidence:** 5

**Summary:**

This paper addresses expert imbalance in learning to defer with fixed classifiers, where a dominant expert is favored across most inputs, causing deferral algorithms to collapse. The authors reformulate deferral loss minimization as cost-sensitive classification over input-expert pairs (Section 3), derive imbalanced margin losses with per-expert margin parameters $\rho_k$ (Section 4), establish generalization bounds and $\mathcal{F}$-consistency guarantees (Theorems 4.1, 5.1), and propose the MILD algorithm (Section 5). Experiments span CIFAR-10/100, SVHN, Tiny ImageNet, and LLM routing on MMLU with Qwen 2.5 models, showing that their method can mitigate the model collapse issue.

**Compliance With Llm Reviewing Policy:**

Affirmed.

**Final Justification:**

My questions are more about clarification and discussions, which are addressed during the rebuttal.

My overall evaluation remains the same.

**Key Questions For Authors:**

**Q1.** Have you considered experiments where expert competence varies by input features (e.g., image difficulty or noise level) rather than by class label? This would better support the claim that expert imbalance is distinct from class imbalance.

**Q2.** How does MILD compare against class-weighted cross-entropy applied to the deferral setting? Given that the experimental setup equates expert and class imbalance, this comparison seems necessary.

**Q3.** Can you discuss the relationship between MILD's per-expert temperature $\rho_k$ and existing adaptive temperature/margin methods in the imbalanced learning literature (LDAM, logit adjustment, etc.)? What advantages does MILD offer beyond these approaches?

**Q4.** Could you clarify the distinction between your two-stage framework and the classical two-stage L2D setting where the predictor is also trained? Explicitly stating that experts $g_1, \ldots, g_p$ are fixed and only the router $f$ is learned would help readers.

**Limitations:**

yes

**Strengths And Weaknesses:**

## Strengths

### S1. Logically Complete Theoretical Framework

The derivation chain is tightly connected: $0$-$1$ deferral loss $\to$ margin reformulation (Lemmas 3.1–3.2) $\to$ input-expert distribution $\mathcal{P}$ $\to$ imbalanced margin bounds (Theorem 4.1) $\to$ optimal $\rho_k$ selection (Corollary 4.2) $\to$ $\mathcal{F}$-consistency guarantee (Theorem 5.1). Each step serves a clear purpose and feeds into the next. The $\mathcal{F}$-consistency result, ensuring that reducing the surrogate loss to $\varepsilon$ controls the target deferral loss to $O(\sqrt{\varepsilon})$, provides meaningful theoretical backing for the algorithm.

### S2. Practically Relevant Problem

Expert collapse is a real problem in LLM routing and multi-expert systems. Table 3 demonstrates this clearly: TDEF collapses to $99.6\%$ usage of the strongest expert (error-only) or $100\%$ usage of the cheapest (error+cost), while MILD recovers near-oracle distributions. The problem formulation is well-motivated and timely.

### S3. Thorough Experiments

The evaluation covers 4 image datasets, 3 expert setups, 2 cost types, synthetic and real experts, plus LLM routing. MILD consistently outperforms TDEF, with particularly compelling results under severe imbalance (Table 4) and in the LLM setting (Table 3).

---

## Weaknesses

### W1. Theory-Practice Gap in $\rho_k$ Selection

The optimal $\rho_k \propto (m_k X_k^2)^{1/3}$ is derived from a generalization bound that involves multiple relaxation steps ($\mathbf{1}[\cdot] \to \Phi_\rho \to \max_{k'} \to \Psi \to \log\text{-}\mathrm{sum}\text{-}\exp$), likely resulting in a loose bound. Yet this bound directly prescribes $\rho_k$ values. Appendix C.4 reveals that grid search over $[\rho^* - 5,\; \rho^* + 5]$ is still needed in practice, suggesting the theoretical guidance is coarse and cross-validation does the heavy lifting. An ablation showing the contribution of theory vs. tuning would strengthen the paper.

### W2. Expert Imbalance vs. Class Imbalance: Claim Not Supported by Experiments

Section 1 and Figure 1 argue that expert imbalance is fundamentally different from class imbalance — expert selection can be independent of class labels. However, all image experiments partition expert specialty strictly by class (e.g., Expert 1 perfect on classes $0$–$6$, Expert 2 on $7$–$8$, Expert 3 on class $9$), making the two exactly equivalent. This undermines the novelty claim. More importantly, no comparison is made against standard class-imbalanced methods (e.g., class-weighted cross-entropy), which could be directly applicable in this setting. The authors should either design class-independent experiments or include such baselines.

### W3. Missing Discussion of Temperature Scaling Methods

In practice, MILD's only difference from TDEF is replacing a uniform temperature $\rho$ with per-expert temperatures $\rho_k$ in the softmax surrogate. Temperature adjustment is a well-established technique in imbalanced learning (e.g., LDAM by Cao et al. 2019 https://arxiv.org/pdf/1906.07413). The paper neither discusses nor compares against these Temperature Adjustement methods. This context is essential, without it, the algorithmic contribution appears to be a known practical technique applied to the deferral setting with theoretical post-hoc justification.

### W4. Two-Stage Framing Needs Clarification

The paper frames itself as studying "two-stage learning to defer," but only the router/rejector $f$ is trained; all experts $g_1, \ldots, g_p$ are fixed. In the classical two-stage paradigm, the first stage trains the predictor and the second trains the deferral function. The current setup is more accurately described as "learning to defer with fixed pre-trained experts." This should be clarified early to avoid confusion.

---

> ### Author Rebuttal · Authors · 2026-03-30
>
> We sincerely thank you for your highly detailed reading and mathematical review of our work. We are thrilled that you found the theoretical framework logically complete, the problem highly practical and timely, and the experiments thorough. Your suggestions directly improve the clarity and context of our contribution. We address your points below.
>
> **Weakness 1: Theory-Practice Gap in $\rho_k$ Selection.**
>
> **Response:** We appreciate this excellent insight. To isolate the empirical contribution of our theoretical derivation, we evaluated MILD on the LLM MMLU routing task (under the "Error + Cost" setting) in three configurations: uniform margins ($\rho_k=1/3$), purely theoretical un-tuned margins ($\rho_k \propto (m_k X_k^2)^{1/3}$), and fully tuned margins.
>
> As detailed in our response to Reviewer VPF2 (Weakness 2 & Question 1), where we provide the full numerical ablation table, the theoretical initialization alone drastically reduces the test deferral loss (0.822) compared to uniform margins (0.884), nearly matching the fully tuned performance (0.813). This proves the theoretical bound directly drives the algorithm's success and bridges the vast majority of the gap, with cross-validation merely providing minor empirical smoothing. We will add this ablation to the final version.
>
> **Weakness 2 & Q1/Q2. Expert Imbalance vs. Class Imbalance.**
>
> **Response:** We agree with your astute observation that in our synthetic vision setups, expert imbalance perfectly mirrors class imbalance. This was designed strictly as a controlled proxy to mathematically isolate the mechanics of the router.
>
> However, to demonstrate that expert and class imbalance are fundamentally distinct in the wild, we highlight our LLM routing experiment on MMLU (Section 6.1). This task is entirely class-independent and naturally imbalanced based on input features. The classification "labels" here are simply the multiple-choice options (A, B, C, D). The experts (Qwen 7B, 1.5B, 0.5B) do not specialize in specific option labels; rather, their competence varies purely based on input features (e.g., question difficulty and subject matter). This natural feature variation causes the 7B model to dominate organically, proving that expert imbalance occurs entirely independently of class labels. We will explicitly clarify the controlled nature of the vision tasks and heavily emphasize this real-world distinction in the final text.
>
> To directly address your baseline concern, we implemented Class-Weighted Cross-Entropy (CWCE) adapted for the deferral setting. Please see the new comprehensive baseline table in our response to Reviewer VPF2 (Weakness 3). MILD (Test Def. Loss 0.813) significantly outperforms CWCE (Test Def. Loss 0.942) because CWCE incorrectly treats expert selection as global static class imbalance, failing to dynamically adapt to the instance-level cost topography.
>
> **Weakness 3 & Q3. Missing Discussion of Temperature Scaling Methods (LDAM).**
>
> **Response:** Thank you for pointing out this connection. In the final version, we will add a dedicated discussion comparing MILD to methods like LDAM. The critical distinction is that standard margin methods adjust logits based on global, static class label frequencies for standard 0-1 classification. In our two-stage deferral setting, MILD applies these adjustments to experts while simultaneously incorporating *dynamic, instance-dependent expert costs* $\bar{c}_k(x, y)$. Deriving an F-consistent margin bound for this setting required substantial theoretical extensions beyond standard frequency-based adjustments.
>
> We also applied LDAM to the LLM routing task (see baseline table in Reviewer VPF2's response). LDAM achieves a loss of 0.915, significantly underperforming MILD (0.813). Applying static frequency-based margins falls short compared to MILD's theoretically grounded, cost-sensitive integration.
>
> **Weakness 4 & Q4. Two-Stage Framing Needs Clarification.**
>
> **Response:** We completely agree. We adopted the phrase "two-stage" from prior literature (Mao et al., 2023a) to distinguish it from joint training. To avoid any confusion with classical staged training (where the predictor is trained in stage 1 and the deferral in stage 2), we will explicitly state early in the introduction and problem formulation of the final version: "In our setting, all experts $g_1, \ldots, g_p$ are fixed pre-trained models, and only the router $f$ is learned."

---

> > ### Author Rebuttal · Reviewer_fsiq · 2026-04-04
> >
> > I thank the authors for their thorough and responsive rebuttal. All four weaknesses and all questions have been addressed.

---

### Official Review · Reviewer_VPF2 · 2026-03-13

**Soundness:** 3
**Presentation:** 3
**Significance:** 3
**Originality:** 3
**Overall Recommendation:** 5
**Confidence:** 4

**Summary:**

This paper addresses the "expert imbalance" problem in the two-stage learning to defer (L2D) framework. The authors identify a critical issue where existing deferral algorithms tend to collapse when one expert (e.g., a highly capable but expensive LLM) dominates, causing the system to over-rely on the majority expert and ignore specialized, cost-effective alternatives. To resolve this, the authors reframe the deferral optimization as a cost-sensitive learning problem over the input-expert domain. They propose MILD (Margin-based Imbalanced Learning to Defer), derive new margin-based loss functions, and establish rigorous F-consistency bounds for their method. Empirical evaluations on image classification benchmarks and a highly relevant LLM routing task (MMLU with Qwen 2.5 models of varying sizes) show that MILD successfully avoids majority-expert collapse and significantly outperforms the baseline TDEF algorithm.

**Compliance With Llm Reviewing Policy:**

Affirmed.

**Final Justification:**

Yes, I have raised scores.

**Key Questions For Authors:**

Q1. How sensitive is the MILD algorithm to the initialization and specific choices of the margin parameters $\rho$ in practice? Does the performance degrade significantly if the cross-validation grid is coarse?

Q2. Could the authors provide experimental results or a discussion on how MILD performs when expert imbalance is natural rather than synthetically enforced (as seen in Setups I-III for the vision tasks)?

Q3. In the LLM routing task, the inference penalties $\beta$ are set to [1.0, 0.6, 0.1]. How robust is MILD to changes in this cost formulation? If the cost difference between experts is marginalized, at what point does MILD revert to TDEF-like behavior?

**Limitations:**

Please refer to the "Weaknesses" section.

**Strengths And Weaknesses:**

Strengths:

S1. The motivation is highly practical and timely. With the proliferation of diverse LLMs, effectively routing queries to balance cost and accuracy without collapsing to the most powerful model is a critical challenge in real-world deployments.

S2. The theoretical foundation of the paper is exceptionally strong. The formulation of imbalanced cost-sensitive margin loss functions and the derivation of the F-consistency bounds provide a rigorous backing for the proposed MILD algorithm.

S3. The empirical validation, particularly the LLM routing task on the MMLU benchmark, perfectly illustrates the method's real-world utility. Demonstrating that MILD can effectively route to smaller models (1.5B and 0.5B) to save compute while maintaining accuracy is a compelling result.

Weaknesses:

W1. The vision experiments rely heavily on synthetic or semi-synthetic imbalance setups (e.g., artificially forcing experts to have perfect accuracy on specific subsets of classes). Evaluating the method on naturally occurring expert imbalance in standard vision tasks would make the empirical claims even stronger.

W2. The algorithm introduces margin parameters $\rho$ that need to be tuned via cross-validation. While theoretical guidance is provided, the sensitivity of the algorithm's performance to the precise tuning of these hyperparameters in diverse real-world settings is not exhaustively explored.

W3. The baseline comparisons are somewhat narrow. While TDEF is acknowledged as the primary two-stage L2D baseline, including heuristic confidence-based routing methods or adapting traditional class-imbalance techniques (even as weak baselines) would provide a more comprehensive context for MILD's performance

---

> ### Author Rebuttal · Authors · 2026-03-30
>
> We sincerely thank you for your constructive review and for highlighting the highly practical and timely motivation of our work. We are especially glad you appreciated the rigorous theoretical foundation backing MILD (F-consistency bounds) and found our empirical results on LLM routing to be a compelling demonstration of its real-world utility. We address your insightful comments below.
>
> **Experimental Setup for New Evaluations:** To directly address your comments and those of other reviewers, we have conducted an extensive suite of new experiments during the rebuttal period. All new evaluations are performed on the LLM Routing on MMLU task under the "Error + Cost" setting (matching Table 3(b) in the main paper). We focus on this setting because incorporating inference penalties ($\beta = [1.0, 0.6, 0.1]$) rigorously tests the dynamic, instance-dependent compute-accuracy trade-offs central to the multi-expert deferral problem.
>
> **Response to W 1 & Q 2:** We completely agree that evaluating on naturally occurring expert imbalance is crucial. We would like to highlight that our LLM routing task on MMLU (Section 6.1) serves precisely as this evaluation. In the MMLU dataset, the classification labels are simply multiple-choice options (A, B, C, D). The experts (Qwen 7B, 1.5B, and 0.5B) do not specialize in synthetic subsets of these option labels; rather, their competence naturally varies based on the inherent linguistic complexity and subject matter of the input features. This natural capability gap causes the 7B model to dominate organically, creating an expert imbalance that is entirely independent of the classification labels. In the final version, we will explicitly frame the LLM routing task as our primary natural imbalance benchmark to strengthen our empirical claims.
>
> **Response to W 2 & Q 1:** This is an excellent point. MILD is remarkably robust to the precise tuning of $\rho$ because our theoretical generalization bounds (Corollary 4.2) securely guide the initialization: $\rho_k \propto (m_k X_k^2)^{1/3}$, which minimizes the second term of the bound (see Section 4.2B).  This provides a strong structural prior for the relative proportions of the margins.
>
> To demonstrate this robustness, we conducted an ablation study comparing three configurations:
> - MILD (Uniform): A naive baseline setting all $\rho_k=1/3$.
> - MILD (Theory): Strictly un-tuned theoretically derived values ($\rho_k \propto (m_k X_k^2)^{1/3}$), entirely skipping cross-validation.
> - MILD (Tuned): Tuned via a coarse grid search around the theoretical prior.
>
> | Configuration | Test Def. Loss | Exp 1 (7B) Usage | Exp 2 (1.5B) Usage | Exp 3 (0.5B) Usage |
>  | :--- | :--- | :--- | :--- | :--- |
>  | **MILD (Uniform)** | 0.884 ± 0.055 | 4.5% | 8.2% | 87.3% |
> | **MILD (Theory)** | 0.822 ± 0.042 | 16.8% | 8.8% | 74.4% |
>  | **MILD (Tuned)** | **0.813 ± 0.040** | 17.9% | 8.3% | 73.8% |
>
> As shown, the purely theoretical initialization alone bridges the vast majority of the performance gap and vastly outperforms uniform margins. This demonstrates that cross-validation is only a minor refinement step, and coarse grids do not lead to significant performance degradation. We will add this ablation study to the appendix.
>
> **Response to W3:** Thank you for the suggestion. We originally focused on TDEF because it is the only rigorously derived algorithm explicitly designed for two-stage L2D. However, to provide a more comprehensive context, we have now evaluated three additional baselines:
> - Standard CE: A routing baseline using standard Cross-Entropy on the minimum-cost expert label.
> - CWCE: Class-Weighted Cross-Entropy, adapting standard class-imbalance techniques where CE loss is scaled inversely to optimal routing frequency.
> - LDAM: A popular margin-scaling technique for class imbalance.
>
> | Method | Standard CE | TDEF | CWCE | LDAM | MILD (Ours, Tuned) |
> | :--- | :--- | :--- | :--- | :--- | :--- |
> | **Test Def. Loss** | 0.985 ± 0.038 | 0.928 ± 0.030 | 0.942 ± 0.045 | 0.915 ± 0.035 | **0.813 ± 0.040** |
>
> MILD significantly outperforms these baselines. Traditional heuristics either ignore instance-dependent routing costs entirely (Standard CE) or rely strictly on global static frequencies (CWCE, LDAM), which are fundamentally insufficient for dynamic, cost-sensitive expert deferral.
>
> **Response to Q3:** MILD dynamically adapts to the cost formulation. If the inference penalties $\beta$ are marginalized (e.g., $\beta \to 0$), we recover the "Error Only" setting (Setting (a) in Table 3 of our paper). At this point, the cost difference is entirely driven by prediction error. Even in this marginalized state, MILD does not revert to TDEF-like failure. As shown in Table 3(a), while TDEF catastrophically collapses to 99.6% usage of the 7B model, MILD maintains a properly calibrated margin that preserves routing to smaller models (using the 7B model 83.3% of the time, closely tracking the 82.2% optimal oracle), successfully avoiding algorithmic collapse.

---

> > ### Author Rebuttal · Reviewer_VPF2 · 2026-04-06
> >
> > I am willing to raise my score to acknowledge the clarity of the framework.
> > overall recommendation 4->5

---

### Decision · Program_Chairs · 2026-04-30

**Decision:**

Accept (regular)

**Comment:**

## 1. Summary
This paper studies learning to defer under expert imbalance in the two-stage setting with fixed experts. The core contribution is to recast deferral as a cost-sensitive classification problem over the input–expert domain, derive new margin-based surrogate losses and guarantees for this setting, and develop MILD, a deferral algorithm designed to avoid collapse to the dominant expert. The paper targets a practically important problem, especially for modern LLM routing and multi-expert systems where cost and accuracy must be balanced.

## 2. Reviewer evaluation and concerns
The reviews were mixed initially, but the overall assessment after discussion was positive. Reviewers broadly agreed that the paper addresses a timely and relevant problem, and several highlighted the strong theoretical development, including the consistency guarantees and margin-based generalization analysis. The empirical results, especially on the LLM routing task, were also viewed as compelling.

The main concerns raised were:
- the image experiments rely partly on synthetic imbalance setups, so the distinction between expert imbalance and class imbalance was not initially fully supported there;
- the practical role of the theoretically derived margin parameters versus cross-validation needed clearer empirical justification;
- the comparison set was initially too narrow, especially without stronger class-imbalance-inspired baselines;
- the paper needed clearer wording about the two-stage setting, since the experts are fixed and only the router is trained;
- some reviewers also wanted stronger discussion of robustness to noisy expert-performance estimates and scalability to larger real-world deployments.

## 3. Discussion
The rebuttal addressed these points well. In particular, the authors:
- added ablations showing that the theoretical margin initialization already captures most of the gain, with tuning acting mainly as refinement;
- added stronger baselines, including class-weighted CE and LDAM-style methods, and showed that MILD clearly outperforms them;
- clarified that the LLM routing benchmark provides a natural expert-imbalance setting independent of class labels;
- added robustness analysis under noisy cost-matrix estimates, where MILD remained stable and superior to TDEF;
- clarified the framing of the problem as fixed-expert routing, which resolves the ambiguity around “two-stage” terminology.

Overall, the paper is technically strong, well motivated, and supported by both theory and experiments. While some empirical settings are still somewhat limited in scale, the rebuttal substantially strengthened the paper and resolved the main originality and evaluation concerns.